# Mammalian APE1 controls miRNA processing and its interactome is linked to cancer RNA metabolism

Giulia Antoniali [1], Fabrizio Serra[1,8], Lisa Lirussi[1,9], Mikiei Tanaka[2], Chiara D'Ambrosio[3], Shiheng Zhang[4], Slobodanka Radovic[5], Emiliano Dalla[6], Yari Ciani[6], Andrea Scaloni[3], Mengxia Li[4], Silvano Piazza [6,7] & Gianluca Tell [1]

Mammalian apurinic/apyrimidinic endonuclease 1 is a DNA repair enzyme involved in genome stability and expression of genes involved in oxidative stress responses, tumor progression and chemoresistance. However, the molecular mechanisms underlying the role of apurinic/apyrimidinic endonuclease 1 in these processes are still unclear. Recent findings point to a novel role of apurinic/apyrimidinic endonuclease 1 in RNA metabolism. Through the characterization of the interactomes of apurinic/apyrimidinic endonuclease 1 with RNA and other proteins, we demonstrate here a role for apurinic/apyrimidinic endonuclease 1 in pri-miRNA processing and stability via association with the DROSHA-processing complex during genotoxic stress. We also show that endonuclease activity of apurinic/apyrimidinic endonuclease 1 is required for the processing of miR-221/222 in regulating expression of the tumor suppressor PTEN. Analysis of a cohort of different cancers supports the relevance of our findings for tumor biology. We also show that apurinic/apyrimidinic endonuclease 1 participates in RNA-interactomes and protein-interactomes involved in cancer development, thus indicating an unsuspected post-transcriptional effect on cancer genes.

[1] Department of Medicine, Laboratory of Molecular Biology and DNA repair, University of Udine, p.le M. Kolbe 4, Udine 33100, Italy. [2] Laboratory of Biochemistry, National Heart Lung and Blood Institute, National Institutes of Health, 50 South Drive, MSC-8012, Bethesda, MD 20892-8012, USA. [3] Proteomics and Mass Spectrometry Laboratory, Institute for the Animal Production System in the Mediterranean Environment (ISPAAM) National Research Council (CNR) of Italy, via Argine 1085, Naples 80147, Italy. [4] Cancer Center of Daping Hospital, Third Military Medical University, Chongqing 400042, China. [5] IGA Technology Services srl, via J. Linussio 51, Udine 33100, Italy. [6] Laboratorio Nazionale CIB, Area Science Park Padriciano, Trieste 34149, Italy. [7] Bioinformatics Core Facility, Centre for Integrative Biology, CIBIO, University of Trento, via Sommarive 18, Povo, Trento, TN 38123, Italy. [8] Present address: Clinical and Experimental Onco-Hematology Unit, Centro di Riferimento Oncologico, I.R.C.C.S., via Franco Gallini 2, Aviano (PN) 33081, Italy. [9] Present address: Department of Clinical Molecular Biology, University of Oslo and Akershus University Hospital, Sykehusveien 27, Nordbyhagen 1474, Norway. Correspondence and requests for materials should be addressed to M.L. (email: mengxia.li@outlook.com) or to S.P. (email: silvano.piazza@unitn.it) or to G.T. (email: gianluca.tell@uniud.it)

The human apurinic/apyrimidinic endonuclease 1 (APE1) is a multifunctional DNA repair protein belonging to the base excision repair (BER) pathway. APE1 also plays non-repair roles in the regulation of the expression of human genes during oxidative stress[1]. Besides filling a crucial role in the maintenance of genome stability, APE1 also acts as a master regulator of the cellular response to genotoxic damage via direct and indirect mechanisms. We recently characterized a direct role of APE1 in the transcription of the SIRT1 gene through the binding of nCaRE-sequences present on its promoter, demonstrating that BER-mediated DNA repair may promote the initiation of transcription of the SIRT1 gene in response to oxidative DNA damage[2]. APE1 may also influence the onset of inflammatory and metastatic progression through its redox-mediated stimulation of DNA-binding activity of numerous transcription factors[3] regulating cancer-related genes. Importantly, by regulating expression of the multidrug resistance gene *MDR1*[4, 5] and the phosphatase and tensin homolog (PTEN) tumor suppressor[6, 7], APE1 has been implicated in chemoresistance. Only recently has it been hypothesized that this protein may play an unsuspected function in RNA metabolism affecting gene expression[8–10]. Therefore, studying overlooked roles of APE1 addresses several unsolved questions in cancer biology. Accordingly, we and others have started to unravel the molecular involvement of APE1 in RNA processing[11–13].

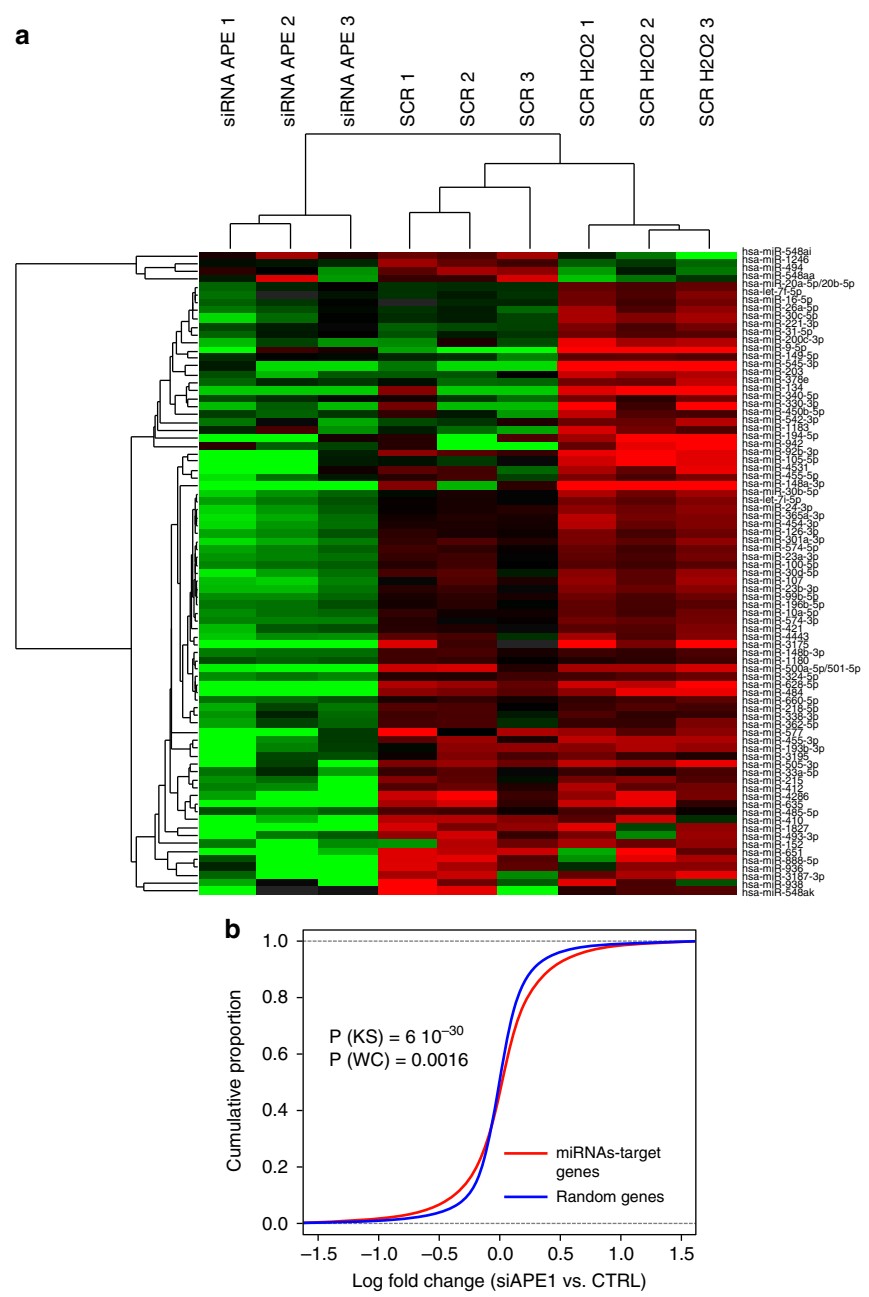

**Fig. 1** miRNA profiling of H$_2$O$_2$-treated and APE1-knocked down HeLa cells. **a** Hierarchical-clustering analysis showing miRNAs differentially expressed in HeLa cell clones stably transfected with scrambled siRNA control (*SCR*), with an APE1 siRNA (*siAPE*) or SCR after stimulation with 1 mM H$_2$O$_2$ for 15 min. The heatmap diagram shows the centered miRNA expression values in logarithmic scale across the three groups of samples. **b** Empirical cumulative distribution function (*ECDF*) curves for expression changes (log fold change) of miRNAs-target genes (*red line*) vs. those of random mRNAs (*blue line*). Statistical significance of the difference between ECDFs is indicated (*P*-value from Kolmogorov–Smirnov (*KS*) test and Wilcoxon (*WC*))

**Table 1 The most enriched canonical pathways associated with the target genes of miRNAs modulated upon APE1-silencing**

| Ingenuity canonical pathways | −log(P-value) | Ratio |
|---|---|---|
| Molecular mechanisms of cancer | 8.39 | 0.12 |
| Chronic myeloid leukemia signaling | 6.61 | 0.18 |
| Cardiac hypertrophy signaling | 6.36 | 0.13 |
| Glioma signaling | 6.22 | 0.17 |
| Estrogen-mediated S-phase entry | 6.20 | 0.38 |
| Glioblastoma multiforme signaling | 5.96 | 0.15 |
| IL-8 signaling | 5.87 | 0.13 |
| PPARa/RXRa activation | 5.63 | 0.14 |
| Endothelin-1 signaling | 5.25 | 0.13 |
| PEDF signaling | 5.22 | 0.18 |
| Myc-mediated apoptosis signaling | 4.80 | 0.19 |
| HGF signaling | 4.69 | 0.15 |
| Glucocorticoid receptor signaling | 4.59 | 0.11 |
| Pancreatic adenocarcinoma signaling | 4.55 | 0.14 |
| Hepatic fibrosis/hepatic stellate cell activation | 4.42 | 0.12 |
| G-protein coupled receptor signaling | 4.26 | 0.10 |
| Role of NFAT in cardiac hypertrophy | 4.11 | 0.12 |

Statistical significance and the ratio of genes included in the pathway are shown. Target genes prediction and pathway enrichment analysis were performed using IPA (see also Supplementary Data File 1). *APE1* apurinic/apyrimidinic endonuclease 1, *HGF* hepatocyte growth factor, *IPA* Ingenuity Pathway Analysis, *PEDF* pigment epithelium-derived factor, *NFAT* nuclear factor of activated T cells

The above-mentioned functions of APE1 are modulated through interactions with several protein partners, some involved in ribosome biogenesis and RNA processing (e.g., nucleophosmin 1 (NPM1) and nucleolin (NCL))[14]. Disruption of this interaction network might impair BER function[9, 15], as we recently demonstrated in acute myeloid leukemia (AML) cells expressing a mutated form of the nucleolar protein NPM1[16]. In that and additional works, we and others showed that APE1: (i) binds, in vitro, structured RNA molecules via its 33 amino acids N-terminal domain[17]; (ii) cleaves abasic single-stranded RNA, taking part in RNA-decay dependent on its endonuclease activity; (iii) has 3′-RNA phosphatase and 3′-exoribonuclease activities; (iv) regulates c-Myc mRNA levels and half-life in tumor cells[18]. Interestingly, no other known enzyme seems to be devoted to the removal of abasic or oxidized RNA or the removal of 3′-phosphates of RNA molecules.

Damage to RNA such as oxidation or abasic site formation, may have profound effects on gene expression and is emerging as a common feature in different human pathologies, including cancer[19]. Oxidized RNA[20] or RNA-containing abasic sites[21] show inhibitory effects on reverse transcriptase activity, and oxidized mRNA[22] or mRNA with abasic sites[23] exhibit compromised translation activity and fidelity[24]. Oxidation of miRNAs may also regulate cellular events by modulating their effects on the specific target gene[25]. Therefore, control of the processing and decay of miRNAs during genotoxic stress may represent an important mechanism of chemoresistance in cancer cells.

By using APE1 knockdown models, we and others have demonstrated the pleiotropic ability of this protein to regulate the expression of hundreds of genes associated with cancer cell proliferation, invasion and chemoresistance[14, 26]. Possible changes in miRNA processing underlying these effects have not generally been investigated. Interestingly, in AML cells expressing cytoplasmic NPM1 (NPM1c+) that alters APE1 endonuclease function and intracellular location[16], there is dysregulation of miR-221/222 processing[27]. Strikingly, since it was previously demonstrated that miR-221/222 regulate the expression of the oncosuppressor PTEN[28–30], a plausible hypothesis could be that

APE1 functional dysregulation may impact on gene expression through miRNome regulation. An exhaustive list of target genes (i.e., RNAs and mRNAs), miRNAs and ncRNAs directly regulated by APE1 during cell response to genotoxic treatment and that may specifically mediate cancer cell resistance to chemotherapy is still lacking to date.

The present work aims at testing whether APE1 may indirectly regulate gene expression through post-transcriptional mechanisms involving miRNAs processing and/or RNA regulation. Altogether, our data provide major insights into APE1-regulated transcriptome and APE1-regulated interactome, and suggest that APE1–miRNA processing, under genotoxic stress conditions, may represent a new paradigm of miRNA regulation in cancer biology with relevance on chemoresistance.

## Results

**miRNA profiling of $H_2O_2$-treated and APE1-depleted HeLa cells.** We hypothesized that APE1 modulates a cellular response to oxidative stress through post-transcriptional regulation of miRNA expression. This hypothesis was tested in two ways. We first tried to identify whether miRNAs regulated by early times of $H_2O_2$-treatment are involved in the regulation of PTEN, a known APE1 target gene[6, 7, 29, 30]. We also evaluated whether APE1-knockdown (APE1-kd) was associated with a significant alteration in the overall miRNome profile. We did not combine both conditions (i.e., oxidative stress and APE1-kd) in order to avoid selection of off-targets effects due to non-specific triggering of DNA damage response (DDR) by simultaneously exposing the cells to oxidative damage in a context of BER-deficiency. For the first test, we performed a high-throughput miRNA expression analysis of HeLa cells upon acute oxidative stress. For the second, we studied the effect of APE1-silencing on the overall miRNome profile. We used HeLa cell clones stably transfected with: (i) scrambled siRNA control (SCR), treated or not with 1 mM $H_2O_2$ for 15 min; and (ii) inducible APE1-specific siRNA (siAPE1 or APE1-kd) cells[14] (Fig. 1a, Supplementary Table 1 and Supplementary Fig. 1a).

The NanoString nCounter Human v2 miRNA Expression platform was used to obtain miRNome profiles. By using a hierarchical-clustering approach and principal components analysis, we were able to confirm the good reproducibility of the data obtained in the biological replicates (Supplementary Fig. 1b, c).

The comparison between the SCR upon $H_2O_2$-treatment and the SCR alone showed 26 upregulated (e.g., miR-221-3p, miR-134, and miR-9-5p) and four downregulated miRNAs (i.e., miR-494, miR-548aa, and miR-548ai) out of 800 total profiled molecules ($\geq 1$ logFC, $q$-value $\leq 0.1$) (Fig. 1 and Supplementary Data File 1). A consistent number of miRNAs was induced by the $H_2O_2$-treatment, in accord with previous studies[25]. Interestingly, miR-221, which targets the PTEN gene,[29, 31] represented an ideal candidate to further test our general hypothesis (see below). In the second set of experiments, APE1-kd downregulated 55 miRNAs (including miR-484, miR-635, and miR-410), with none upregulated, compared to the SCR control (Supplementary Data File 1). The latter result was in line with previous work, in which human osteosarcoma cells (HOS) were transfected with APE1 siRNA[26]. In fact, 4 out of 12 miRNAs that were downregulated in HOS (i.e., let-7i-5p, miR-324-5p, miR-421, and miR-484) were also significantly downregulated in APE1-kd HeLa cells. Four other transcripts showed logFC expression values below the applied cutoff, while the remaining four did not significantly differ. Comparing the expression level variations between the two experiments showed HeLa cells having a significantly higher number of downregulated miRNAs than

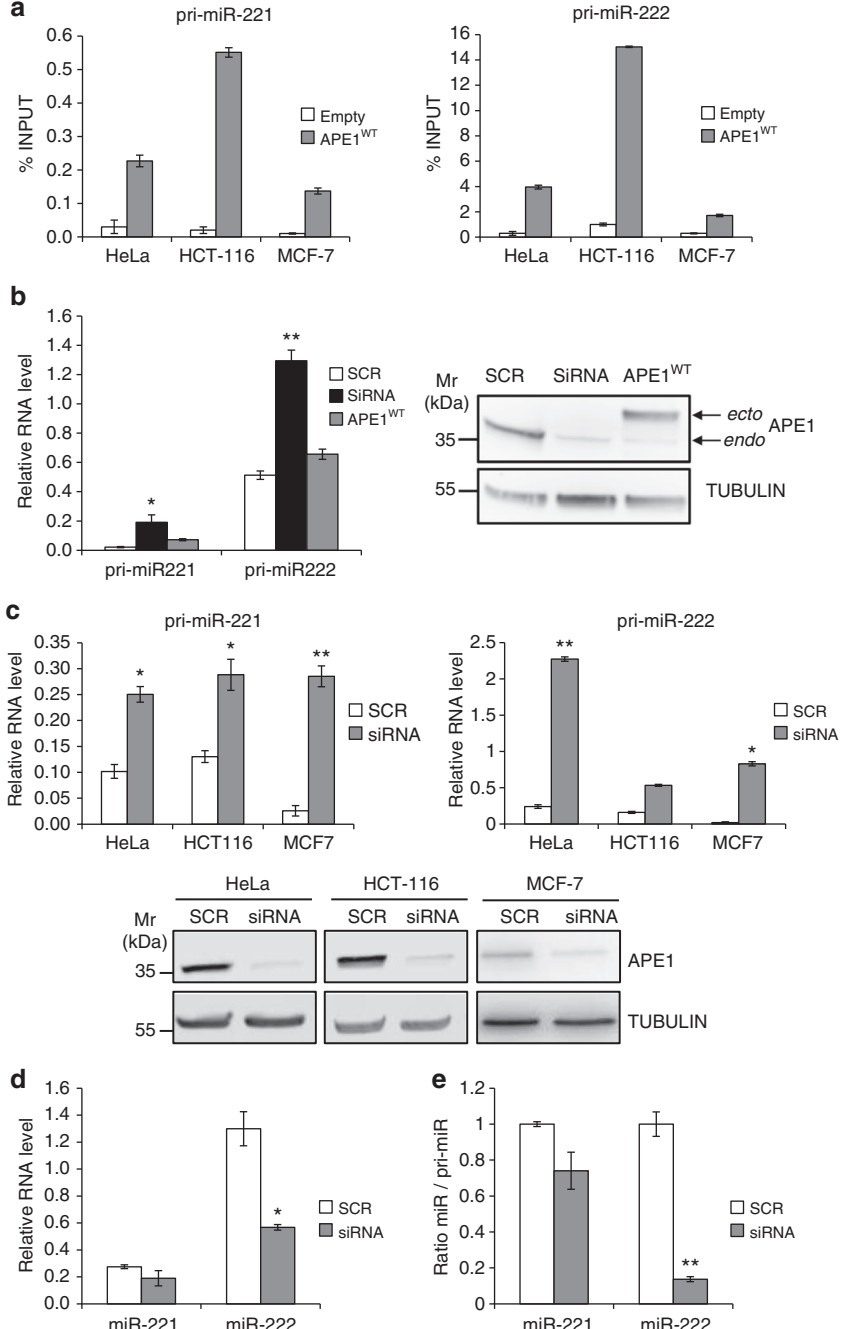

**Fig. 2** APE1 binding to pri-miR-221/222. **a** Validation of APE1 binding to pri-miR-221 and pri-miR-222 in different human cancer cell lines. qRT-PCR of pri-miRs bound by APE1 in different cell lines transfected with either empty vector or with a vector expressing APE1^WT FLAG-tag protein. Data are presented as fold percentage of the amount of immunoprecipitated pri-miR relative to that present in total input RNA. **b** Pri-miR-221 and pri-miR-222 expression levels evaluated by qRT-PCR analysis of HeLa cell clones silenced for APE1 expression. Total RNA was extracted from HeLa cell clones stably transfected with scrambled siRNA control (*SCR*), with an APE1 siRNA (*siRNA*) and reverse transcribed as described in the Methods section. Histograms show the detected levels of pri-miR-221 and pri-miR-221 normalized to GAPDH levels. *Asterisks* represent a significant difference with respect to control (SCR). *$P < 0.05$, **$P < 0.001$, Student's *t*-test. *Right*, western blotting analyses of HeLa cell clone extracts silenced for APE1 expression. **c** Pri-miR-221 and pri-miR-222 expression levels evaluated by qRT-PCR analysis of different cell lines. Total RNA was extracted from HeLa, HCT-116, and MCF-7 cell lines transiently silenced for APE1 and reverse transcribed. Histograms show the detected levels of pri-miR-221 and pri-miR-221 normalized to GAPDH levels. *Asterisks* represent a significant difference with respect to control (SCR). *$P < 0.05$, **$P < 0.001$, Student's *t*-test. *Bottom*, representative western blotting analyses to confirm APE1 silencing in HeLa, HCT-116, and MCF-7 cells. Tubulin was used as loading control. **d** Mature miR-221 and miR-222 expression levels evaluated by qRT-PCR analysis of HeLa cells silenced for APE1 expression. Total RNA was extracted from HeLa cell clones stably transfected with scrambled siRNA control (*SCR*), with an APE1 siRNA (siRNA), and reverse transcribed. Histograms show the detected levels of miR-221 and miR-222 normalized to RNU44 levels. *Asterisks* represent a significant difference with respect to control (SCR). *$P < 0.05$, **$P < 0.001$, Student's *t*-test. **e** Mature miR to pri-miR ratios in HeLa cells clones silenced for APE1 expression. Mature miR-221 and miR-222 were measured by qRT-PCR analysis, normalized to RNU44, and expressed relative to GAPDH-normalized pri-miR-221/222. *Asterisks* represent a significant difference with respect to control (SCR). *$P < 0.05$, **$P < 0.001$, Student's *t*-test

found in HOS cells, with a median logFC expression 4-fold higher than for HOS (−1.8 and −0.4, respectively) (Supplementary Data File 1). We found two miRNAs that were downregulated in both conditions (miR-494 and miR-1246), and two others that were upregulated upon $H_2O_2$-treatment but were downregulated after APE1 silencing (miR-30b-5p, miR-92b-3p).

Target gene prediction and pathway enrichment analysis for the APE1-kd cells, performed using Ingenuity Pathway Analysis (IPA; QIAGEN Bioinformatics), demonstrated significant enrichment for molecular pathways of cancer development associated with miRNA dysregulation (Table 1 and Supplementary Data File 1). To determine whether the downregulation of miRNAs upon APE1 depletion affects mRNA expression, we compared the cumulative changes for genes that are miRNA targets vs. those of random sets of mRNAs. Gene expression data were obtained from a previous investigation from our laboratory[14]. To correct for bias in the random set, we performed 1000 comparisons in which the log(fold change) values were randomly selected from the whole data set, while maintaining the size of the original distribution (Fig. 1b). Using both the Kolmogorov–Smirnov test and Wilcoxon test, the Benjamini and Hochberg method (BH) adjusted $P$-values were statistically significant (with confidence level = 0.95, $P < 6 \times 10^{-30}$ and $P = 0.0016$, respectively; see Methods for further details and Supplementary Data File 1 for the miRNA target prediction table).

Overall, these results suggest a positive impact of APE1 protein on specific miRNA expression levels, possibly acting on the early processing events and allow identifying miR-221 as a candidate for testing, as a "proof of concept", the hypothesis that APE1 regulates the expression of target genes involved in chemoresistance.

**Precursor forms of miR-221/222 are bound by APE1.** We then investigated the molecular mechanism of APE1-affecting miRNA expression, focusing our attention on miR-221 and miR-222, because they are correlated in a polycistronic cluster and relevant for PTEN expression[28, 29, 31]. Due to the ability of APE1 to directly bind structured RNA molecules[11, 12] and the double-stranded nature of pri-miRNAs, we first tested the ability of APE1 to bind the primary transcript (i.e., pri-miRNA) forms of these miRNAs, by performing RNA immunoprecipitation (RIP)-analyses in different cancer cell lines (i.e., HeLa, MCF-7 and HCT-116) upon transient transfection (Fig. 2a). To this end, cell lines were transiently transfected with FLAG-tagged APE1 wild-type protein-encoding plasmid and the immunoprecipitated RNA was analyzed by qRT-PCR to assess the levels of each pri-miR-221/222 bound by APE1. As shown in Fig. 2a, we efficiently immunoprecipitated both pri-miRNAs in all cancer cell lines tested.

Considering the potential of APE1 to regulate miRNA processing via enzymatic cleavage of RNA with secondary structure[11, 12], we investigated the role of APE1 in miR-221/222 processing efficiency. First, we checked if the pri-miR-221/222 expression level was affected by APE1-kd in either HeLa cell clones with a stably transfected siRNA vector (Fig. 2b), or in cells transiently transfected with a different APE1-specific siRNA (Fig. 2c). In both cases, APE1 depletion was followed by an increase in pri-miR-221/222 expression compared to control siRNA. In accord with this result, HeLa cell clones re-expressing wild-type APE1 via an siRNA-resistant mRNA[13] had almost the same level of the two pri-miR transcripts as the cell clone expressing a scrambled vector (SCR) (Fig. 2b).

This increased expression of pri-miR-221/222 in APE1-depleted cells suggested that the primary transcript forms might accumulate due to impairment of the early steps of miRNA-processing dependent on APE1. Therefore, we assessed if the

accumulation of the pri-miR-221/222 was paralleled by a decrease in the expression of mature miR-221/222 upon APE1-silencing (Fig. 2d). miR-222 was ~6-fold more abundant than miR-221 in control (SCR) HeLa cells, which was significantly decreased upon APE1 silencing (siRNA). In the case of miR-221, the changes were not statistically significant. On the other hand, the amount of each pri-miR was increased upon APE1 silencing, as shown by the corresponding miR/pri-miR ratio (Fig. 2e). The APE1 role was confirmed in APE1-ko mouse cells (CH12F3)[32] (Supplementary Fig. 2). Altogether, these results indicate that, although to a different extent possibly dependent on the absolute miR expression levels, the pri-miR-221/222 processing is compromised in APE1-depleted cells.

**Inhibition of APE1 impairs miR-221/222 expression.** To define the role of APE1 in processing miR-221/miR-222 precursors, we tested whether the endonuclease or the redox activities of the protein were involved. For this purpose, we treated HeLa cells with different inhibitors of specific APE1 functions: (i) compound #3, a catalytic inhibitor of APE1 endonuclease activity[33]; (ii) fiduxosin, a recently characterized inhibitor of the APE1/NPM1 interaction[34], which localizes and activates APE1 function in nuclear BER[16]. Thus, fiduxosin inhibits the protein endonuclease activity in cells through a mechanism that is different from that of compound #3, but having a similar extent[34]; (iii) E3330, a well-known inhibitor of APE1 redox activity now used in clinical trials[35, 36]. Cells were challenged with all these APE1 inhibitors for 24 h, and the miR-221/222 precursor and mature forms were quantified through qRT-PCR (Fig. 3a). Time and doses of treatments were chosen based on their effect on cell viability and previous published data[34, 37]. The interference with APE1 endonuclease activity by compound #3 and fiduxosin resulted in an accumulation of pri-miR-221 and pri-miR-222 (Fig. 3a). Conversely, the redox-inhibitor exerted only a slight increase in the amount of mature miR-222. Inhibition of APE1 endonuclease activity was demonstrated by in vitro cleavage assays using a substrate bearing an AP site (Fig. 3a and Supplementary Fig. 3a). Of note, the impaired miRNA processing observed upon APE1 endonuclease inhibition was not associated with any effect on the total amount of APE1 protein (Supplementary Fig. 3b).

To complement the inhibitor experiments, we expressed mutant APE1 proteins with different defects in APE1-kd cells. These included a nuclease-defective form (APE1E96A)[38], a redox-defective form (APE1C65S)[39], and a protein lacking the N-terminal 33 residues (APE1NΔ33) which does not interact with NPM1[13, 40]. These proteins were expressed at comparable levels, while endogenous APE1 was mostly suppressed (Fig. 3b).

The results with the mutant APE1 proteins (Fig. 3b) supported the conclusion that the endonuclease function of APE1 and its N-terminal region are essential for the normal processing of pri-miR-221/222. In contrast, the redox-defective APE1C65S showed a small increase of miR-222 mature form relative to the precursor, which may be explained by secondary effects due to the expression of this mutant in HeLa cells, as we previously described[41]. Notably, since the APE1NΔ33 lacks critical localization signals and has impaired interactions with other proteins besides a reduced interaction with NPM1[14], we cannot exclude that both regulatory aspects could participate in the cellular endpoints measured.

In the OCI/AML3 cell line that stably expresses the aberrantly cytoplasmic NPMc+ mutant protein, APE1 is also mis-localized to the cytoplasm, which impairs nuclear BER[16]. In these cells, we observed an increased accumulation of pri-miR-221/222 (Fig. 3c)

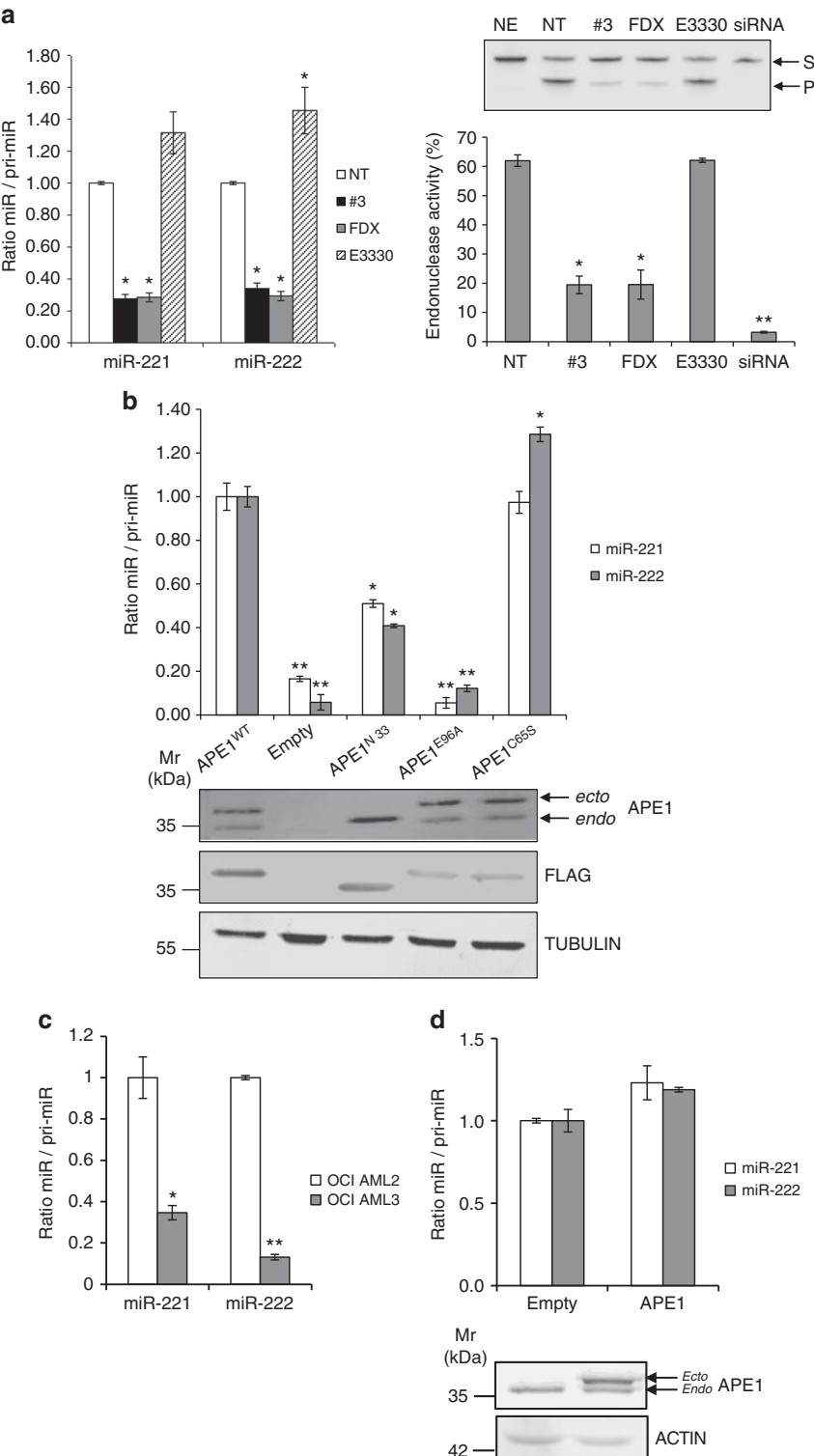

compared to cells with wild-type NPM1. Such an effect was previously reported without a molecular explanation of the results[27]. These data paralleled those obtained with fiduxosin[34] indicating that NPM1 exerts a positive effect on APE1 pri-miRNA-processing activity. As APE1 depletion impaired processing of pri-miR-221 and pri-miR-222, we also tested if APE1 overexpression would give the opposite effect (Fig. 3d). HeLa cells were transfected with a plasmid encoding the APE1–FLAG-

tagged protein, and the ratio of mature miR to pri-miR was evaluated. The absence of a statistically significant effect, suggests that other proteins may be the rate-limiting factors in the pri-miR processing pathway.

Overall, our data show that the endoribonuclease activity of APE1 appears required for the early phases of miR-221/222 processing but that additional protein factors may also play a role.

**APE1 interaction with DROSHA is enhanced by genotoxic damage.** APE1-deficiency leads to increased RNA oxidation[13]. We thus speculated that the APE1 requirement in pri-miR-221/222 processing could reflect an action of APE1 on RNA-decay pathways for pri-miRNAs, possibly associated with the DROSHA microprocessor complex. A possible interaction between APE1 and the DROSHA microprocessor complex was tested using the proximity ligation assay (PLA) in HeLa cells under oxidative stress conditions, which may modulate their interaction. The results in Fig. 4a point to an interaction between APE1 and DROSHA in the nucleoplasmic compartment, where pri-miRNAs are processed under normal conditions. Specificity was demonstrated by the reduced number of PLA spots in a negative control omitting the antibody for DROSHA; as a positive control for an APE1-interacting partner, we confirmed the known interaction between APE1 and NPM1[16, 42] (Supplementary Fig. 4a). Moreover, the lack of detectable interaction between APE1 and the essential microprocessor complex component DGCR8 (DiGeorge critical region 8) or with the auxiliary factor DEAD-box RNA helicase p68 (DDX5) is consistent with a role of APE1 in the early phase of microprocessor pathways (Supplementary Fig. 4b). The APE1/DROSHA interaction was stimulated by $H_2O_2$ at very early times upon treatment, peaking at 15 min of treatment with 1 mM $H_2O_2$, thus supporting the hypothesis for a role of APE1 in the quality control of oxidized RNA[10]. As expected, the $H_2O_2$-induced PLA-signal was greatly reduced in APE1-kd cells compared to cells with the control siRNA (Supplementary Fig. 4c, d). The role of APE1 seems to extend to RNA damaged by non-oxidative agents: treatment with the alkylating compound MMS also stimulated the APE1/DROSHA interaction, though less dramatically than did $H_2O_2$ treatment (Supplementary Fig. 4e).

The interaction between APE1 and DROSHA seems transitory. We were unable to detect any significant interaction between these proteins by Co-IP in experiments whereas the known APE1–NPM1 interaction[13, 42] was easily detected (Supplementary Fig. 4f). This observation was predictable, since the interaction implies the enzymatic activities of two proteins (i.e., APE1 and DROSHA) on RNA molecules with high turnover rates. Alternatively, it may be that the APE1–DROSHA interaction occurs on an RNA molecule that can be degraded during the co-immunoprecipitation (Co-IP) procedure.

Inspecting the pri-miR-221/222 expression levels after a 15 min treatment with 1 mM $H_2O_2$, we observed a time-dependent increase in the levels of both pri-miRs, with respect to untreated (NT) cells (Fig. 4b). This effect was more pronounced for pri-miR-221. However, this oxidant-induced increase did not correlate with an increase in the mature miRNA forms, as seen in the kinetics of the miR:pri-miR-221/222 ratio (Fig. 4b). This is possibly due to a blockage in the maturation process during oxidative stress under this experimental condition (Fig. 4b). The different kinetics observed in the case of the two miRNAs, particularly once starting the release time upon $H_2O_2$-treatment (indicated as time 0 of release), may be ascribed to a different turnover rate of the two miRNAs.

Finally, as APE1 may be involved in the turnover of damaged pri-miRNAs, we measured the extent of oxidative base loss in pri-miRNA-221/222 as a function of APE1 expression using an aldehyde-reactive probe (ARP)[43]. Indeed, APE1-kd was associated with a significant increase in damage to both pri-miRNAs, with re-expression of wild-type APE1 eliminating this effect (Fig. 4c). We thus hypothesize an unanticipated function of APE1 in the microprocessor complex, possibly associated with pri-miRNA-decay mechanisms and affecting the miRNA maturation processes during genotoxic damage.

**APE1 effect on PTEN-pathway correlates with miR-221/222.** We tested the functional relevance of our findings on the biological targets of miR-221/222 by examining the expression of PTEN, a tumor suppressor protein known to be functionally related to APE1 expression[6]. The effect of both APE1 silencing (Fig. 5a) and inhibition (Fig. 5b) were assessed for PTEN mRNA and protein levels. qRT-PCR and western blotting analyses revealed upregulation of PTEN in APE1-kd cells or in cells treated with compound #3, with a concomitant downregulation of the miR/pri-miR-221/222 ratios. As PTEN negatively regulates the AKT pathway by antagonizing PI3K activity by dephosphorylating $PIP_3$[28], we evaluated the phosphorylation of Akt (p-AKT) in APE1-kd cells. Consistent with PTEN upregulation under APE1 silencing, there was a decrease in p-AKT phosphorylation (Fig. 5c). Conversely, APE1 overexpression caused a partial rescue of p-AKT phosphorylation (Fig. 5c). The effect of APE1 on PTEN expression may be linked to the action of the former in miRNAs processing.

**Correlations of APE1 and miR-221/222 with PTENs in cancer specimens.** To determine the significance for human cancer of the correlation between APE1 and miR-221/222 processing PTEN expression, we analyzed a cohort of 94 tissue samples from chemotherapy-naive and radiotherapy-naive patients diagnosed with colorectal cancer, glioblastoma, breast cancer, cervical

**Fig. 3** Inhibition of APE1 endonuclease activity negatively affects miR-221 and miR-222 processing. **a** Mature miR-221 and miR-222 were measured by qRT-PCR in HeLa cells treated with 20 μM compound #3, 40 μM fiduxosin (*FDX*) and 100 μM E3330 for 24 h, respectively. Mature miRNAs were normalized to RNU44 and expressed relative to GAPDH-normalized pri-miR-221/222. *Right*, AP-site incision activity of total cell extracts from HeLa cells treated with the indicated APE1 inhibitors or HeLa cells silenced for APE1 (siRNA). siRNA cell extracts were used as negative control. The histogram indicates the percentage conversion of an AP site-containing DNA substrate (*S*) to the incised product (*P*). Data are expressed as mean ± SD of three technical replicates from two independent assays. A representative image of the denaturing polyacrylamide gel of the enzymatic reactions is shown. *NE* no cell extract, *NT* non-treated cells. Asterisks represent a significant difference with respect to control (NT).*$P < 0.05$, **$P < 0.001$, Student's *t*-test. **b** Mature miR to pri-miR ratios in HeLa cell clones silenced for the endogenous APE1 expression and transiently transfected with expression plasmids for FLAG-tagged, siRNA-resistant APE1 mutants APE1$^{WT}$, APE1$^{NΔ33}$, APE1$^{E96A}$, and APE1$^{C65S}$. Mature miR-221 and miR-222 levels were measured by qRT-PCR analysis, normalized to RNU44, and expressed as relative to GAPDH-normalized pri-miR-221/222. Asterisks represent a significant difference with respect to control (SCR). *$P < 0.05$, **$P < 0.001$, Student's *t*-test. *Below*, western blotting analysis showing HeLa cell clones silenced for endogenous APE1–protein (*endo*) and re-expressing ectopic APE1–FLAG-tagged mutants (*ecto*). **c** miR-221 and miR-222 expression levels evaluated by qRT-PCR analysis of OCI/AML-2 and AML-3 cells lines. OCI/AML2 cells represent the control expressing a wild-type NPM1 protein, which accumulates within nucleoplasm and nucleoli. Histograms show the ratio between mature miRNAs relative to their GAPDH-normalized precursors. Asterisks represent a significant difference with respect to control (OCI/AML-2).*$P < 0.05$, **$P < 0.001$, Student's *t*-test. **d** miR-221 and miR-222 expression levels evaluated by qRT-PCR analysis of HeLa cells overexpressing APE1 by transient transfection of APE1–FLAG-expressing plasmid. Histograms show the ratio between mature miRNAs relative to their GAPDH-normalized precursors. *Below*, western blotting analysis showing HeLa cells transfected with ectopic APE1 FLAG-tagged plasmid

cancer, and non-small cell lung cancer (NSCLC). APE1 and PTEN protein expression were estimated by immunohistochemistry (IHC), and representative images of APE1-high and APE1-low examples for the five cancer types are shown in Fig. 6a. Statistical analysis shows a trend with higher PTEN-high expression in samples scored as "APE1-low" ($r = -0.665$, $P < 0.0001$ Student's $t$-test), consistent with the in vitro studies.

Both the mature and the primary forms of miR-221/222 were quantified using qRT-PCR, and the corresponding miR/pri-miR ratios were calculated (Fig. 6c, d). As shown in the scatter plots, more samples with a high ratio of mature/primary miR-221/222 were associated with higher APE1–protein levels. Statistical analyses further indicated APE1 correlation with miR-221/222 processing (miR-221: $r = 0.92$, $P < 0.0001$; miR-222: $r = -0.649$,

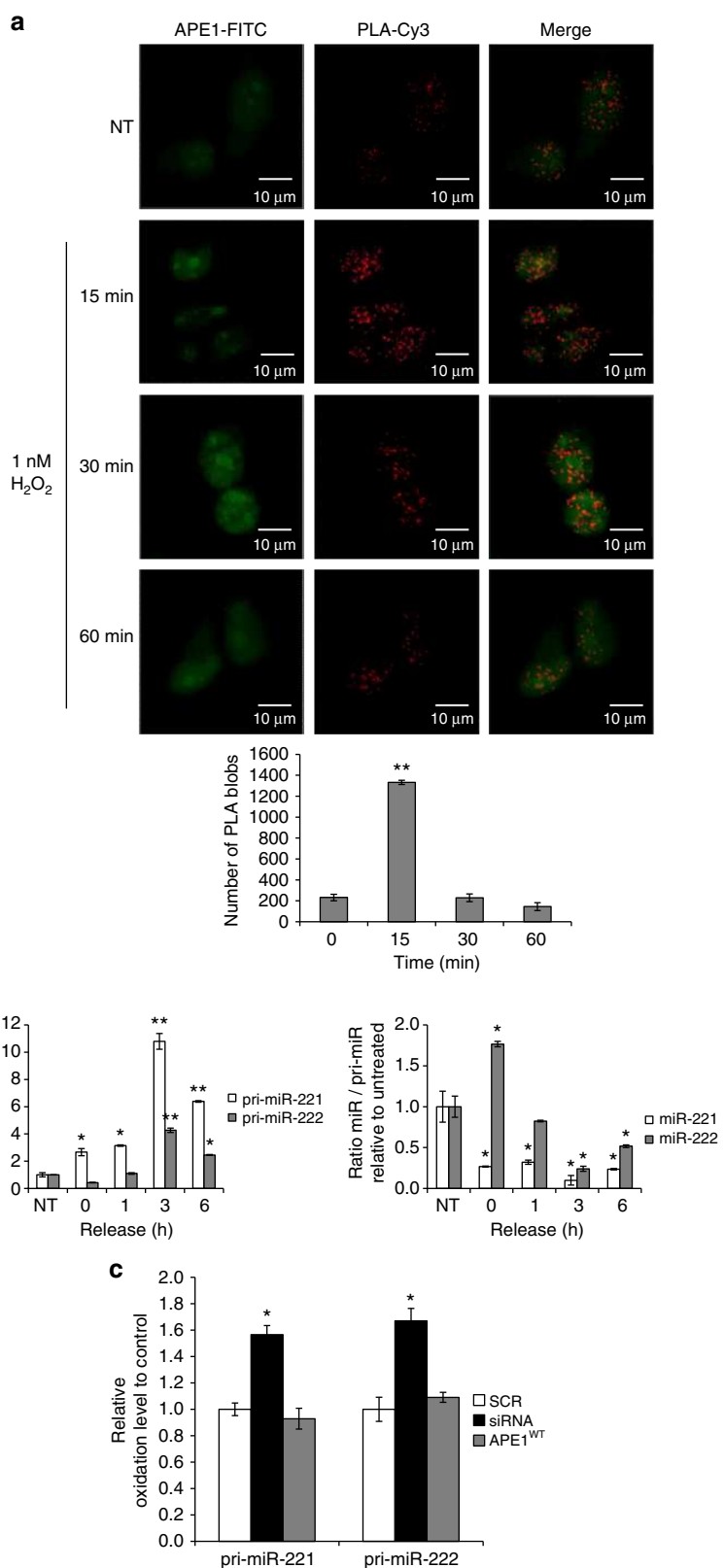

$P < 0.0001$ Student's $t$-test). APE1–protein level was inversely correlated with pri-miR-221/222 (pri-miR-221: $r = -0.491$, $P < 0.0001$ and pri-miR-222: $r = -0.497$, $P < 0.0001$ Student's $t$-test), and with mature miR-221/222 (miR-221: $r = 0.343$, $P < 0.0001$ and miR-222: $r = 0.418$, $P < 0.0001$ Student's $t$-test). Taken together, the relationship between APE1 and miR-221/222 processing, and with PTEN expression, was validated in clinical tumor samples, supporting the hypothesis that the observed mechanism is bona fide a general effect in cancer.

**APE1–protein interactome network dynamics**. The role of APE1 in miRNA processing and its role during genotoxic damage highlight a central function of this protein in RNA metabolism. Therefore, we speculated that APE1 may be a hub involved in a dynamic interaction with many proteins involved in RNA-processing/metabolism as we already observed[10, 13]. To experimentally test this hypothesis, we first implemented the number of known APE1-interacting partners by using proteomic analysis of APE1 co-immunopurified material[3, 13, 14] obtained from whole-cell extracts. This was done with the aim to avoid possible artifacts due to subcellular fractionation procedures and to have a representative interactomic network considering the relative abundance of the different protein species. In this analysis, we took also into account the role of acetylation in modulating the APE1-interactome, because of recent publications demonstrating its association with cancer[44, 45] and for the fact that acetylation is responsible for modulating APE1 subcellular distribution[42] and RNA-binding properties[17, 40] (Supplementary Note, Supplementary Fig. 5a–e, and Supplementary Data Files 2–5). Overall, the APE1-interactome network, characterized in part in our laboratory and in different literature works, actually comprises 103 different protein species including the newly identified ones (Supplementary Data Files 2 and 6). When we functionally annotated this list using IPA, we observed that the majority (93%) of APE1-binding partners were related to five biological pathways and, in particular, 63% of them were linked to processing of RNA (e.g., YB-1, NPM1, RPLP0, NCL, PRPF19), DNA repair (e.g., LIG1, POLB, XRCC1, OGG1, FEN1), and gene expression (e.g., STAT3, NME1, MDM2, TCEB1, POLR3D) (Fig. 7a, b; Supplementary Fig. 6a, b and Supplementary Data File 6).

Moreover, we found that APE1 may undergo acetylation in several residues (i.e., Lys3/6/7/27/31/32/35/141/197/203/227/228) mainly located in the unstructured N-terminus (Supplementary Fig. 5a; and Supplementary Data Files 3 and 4), confirming our previous[40] and literature studies[45], and that acetylation of APE1 is associated to a modulation of its protein interactome network (Supplementary Fig. 5c, d and Supplementary Data Files 3 and 4). Interestingly, the APE1–protein interactome was mediated by RNA molecules. In fact, treatment of immunoprecipitated material with DNase I-free chromatographically purified RNase A mostly reduced the interaction of APE1 with the different interacting proteins, while DNase I-treatment was almost ineffective (Supplementary Fig. 7a). Altogether, these data clearly demonstrate that the APE1–protein interactome network is largely mediated by RNA and is dynamically modulated by acetylation and during genotoxic conditions. Moreover, these findings reinforce the idea that APE1 may act as a multifunctional hub protein, emphasizing the emerging role that APE1 plays in RNA metabolism and the relevance of its protein interactome once considering the many different activities ascribed to this protein in cancer.

**Genome-wide identification of the APE1–RNA-interactome network**. Based on the observation that RNA contributes to the APE1–protein interactome and that APE1 directly binds pri-miRNAs and rRNA[13, 40], we then used an unbiased approach to investigate the associations of APE1 with non-ribosomal RNA species using modified RIP-seq analysis. RNA-bound APE1, extracted from HeLa cell clones expressing an ectopic FLAG-tagged wild-type APE1–protein, was purified using an anti-FLAG antibody whose specificity was already well-characterized in previous IP-studies[2] (Supplementary Fig. 8a). Three independent immunoprecipitation experiments were performed; to further reduce potential false positives, a negative control of resin lacking the proper antibody was also introduced. Input samples for each triplicates were also collected and sequenced. Co-IP Western blot analyses confirmed that FLAG-APE1 was efficiently affinity purified exclusively from HeLa cell extracts immunoprecipitated with the resin carrying the anti-FLAG antibody (Supplementary Fig. 8b). RNA bound by APE1 was then subjected to sequencing analysis and bound transcripts were identified. We obtained an average of 38.84 and 34.23 million reads for the libraries from RIP control cells and APE1-overexpressing HeLa cells, respectively. Among the 1015 RNA molecules, in addition to 989 protein coding genes, we found 26 non-coding elements (2 lincRNAs, 2 ncRNAs, 5 antisense RNAs, 8 pseudogenes, 8 processed transcripts, and 1 miRNA) (Supplementary Data File 7). Since our RNA-seq analyses was not optimized for miRNA/pri-miRNA sequencing, we cannot actually exclude that additional miRNAs/pri-miRNAs could be bound by APE1, as we here demonstrated (Fig. 2a). In order to validate RIP-seq results, among the 1015 predicted RNAs bound by APE1, some RNA targets were also evaluated through qRT-PCR analysis (Supplementary Fig. 8c).

To determine the functions of the APE1-associated-RNA genes (AARGs) by a more global analysis, we investigated for their molecular functions using the Core Analysis function included in IPA. After the analysis, biofunctions and diseases were ordered by the statistical significance score (−log $P$-value). The top five functional annotation clusters of AARGs, considering the biofunctions, are shown in Fig. 8 (see also Supplementary Data File 7). Interestingly, this analysis revealed that the AARGs are mostly involved in RNA-metabolism (transcription, processing,

**Fig. 4** Interaction of APE1 with the DROSHA complex is stimulated by oxidative stress. **a** Nucleoplasmic interaction between APE1 and the DROSHA complex after oxidative stress. HeLa cells were placed on a glass coverslip and treated with 1 mM $H_2O_2$ for 15, 30, and 60 min. PLA reaction was carried out using anti-APE1 and anti-DROSHA antibodies. APE1 expression was detected by using an anti-APE1 antibody and was used as a reference for the nuclei. Data reported in the histogram account for the average number of PLA signals of at least 30 randomly selected cells per condition. **$P < 0.001$, Student's $t$-test. **b** miR-221 and miR-222 expression levels evaluated by qRT-PCR analysis of HeLa cells treated with 1 mM $H_2O_2$ for 15 min and released for 1, 3 or 6 h after treatment. Histograms show the detected levels of pri-miR-221 and pri-miR-221 normalized to GAPDH levels (left) and the ratio between mature miRNAs relative to their GAPDH-normalized precursors (right). Asterisks represent a significant difference with respect to control (NT). NT non-treated. *$P < 0.05$, **$P < 0.001$, Student's $t$-test. **c** Total RNA, isolated from HeLa cell clones, was reacted with aldehyde-reactive probe specifically on oxidative abasic sites, followed by precipitation with magnetic beads. Precipitated oxidized RNA and total RNA were subjected to qRT-PCR individually using TaqMan probe for pri-miR-221 or pri-miR-222. Oxidation levels of miRNAs were determined based on difference in Ct value between oxidized and total RNA. Data are represented as mean ± SD after three replication tests. *$P < 0.05$, Student's $t$-test

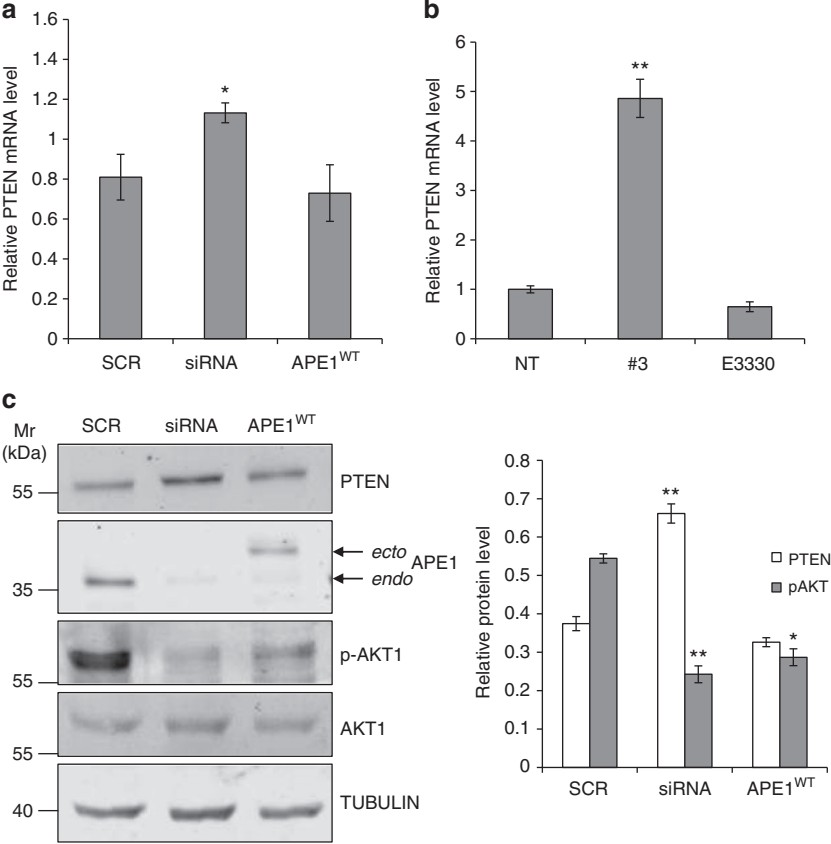

**Fig. 5** Impact of miR-221/222 processing on PTEN protein expression. **a** PTEN mRNA levels evaluated by qRT-PCR analysis of HeLa cell clones silenced for APE1 expression. Total RNA was extracted from HeLa cell clones expressing APE1 WT or APE1 silenced (siRNA) and reverse transcribed. Histogram shows the detected levels of PTEN normalized to GAPDH levels. *Asterisks* represent a significant difference with respect to control (SCR).*$P < 0.05$, Student's *t*-test. **b** PTEN mRNA levels evaluated by qRT-PCR analysis of HeLa cell treated with 20 μM compound #3 or 100 μM E3330 for 24 h, respectively. Histogram shows the detected levels of PTEN normalized to GAPDH levels. *Asterisks* represent a significant difference with respect to control (NT). **$P < 0.001$, Student's *t*-test. **c** PTEN protein level evaluated in HeLa cell clone silenced for APE1 expression. Representative western blotting analyses of total cell extracts of HeLa cell clones. PTEN expression inversely correlates with phosphorylation of Akt1 (*pAkt1*). Histogram reports expression level of PTEN and pAkt protein obtained after quantification of the signal intensity of the corresponding bands. Data represent the means of ± SD of three independent experiments. Tubulin was used as loading control and for data normalization. *Asterisks* represent a significant difference with respect to control (SCR).*$P < 0.05$, **$P < 0.001$, Student's *t*-test

splicing, and oxidation of RNA), supporting our previous hypothesis on the crucial role exerted by APE1 in RNA biology[10–13]. We obtained similar results using DAVID/EASE enrichment analysis[46] (Supplementary Fig. 8d and Supplementary Data File 7). Interestingly, enrichment of AARGs involved in DNA metabolism as well as organization of cytoskeleton and organelles may suggest a potential role of APE1 in RNA-trafficking or RNA-processing events. Moreover, the analysis of diseases clearly showed a central role of APE1–RNA-bound species in various types of cancer (Table 2 and Supplementary Data File 7). Taken together, these analyses show that APE1 binds a highly coherent set of RNA targets, closely related to its roles in both normal biology and disease. Moreover, these findings strongly suggest that one important mechanism, through which APE1 may regulate gene expression, is by directly acting on RNA molecules, possibly through RNA-processing/decay events and involving different protein complexes.

## Discussion

Here, we demonstrated that the BER enzyme APE1 may represent a new hub in RNA-processing events, including miRNA regulation, thus post-transcriptionally affecting gene expression with relevance in chemoresistance. Its association with a network of RNA and protein species, highly related to its functions, opens new perspectives for understanding the multifunctional roles of this unusual DNA-repair enzyme. To the best of our knowledge, this investigation reveals a previously unpredicted function of APE1 in miRNA processing, which may underlie novel key aspects of APE1 in cancer biology. Interestingly, we found that APE1, through direct interaction with the DROSHA microprocessor complex, enhances the post-transcriptional maturation of miR-221/222, thus impacting on PTEN gene expression and affecting the Akt pathway under basal conditions. The observation that oxidative or alkylating agents promote APE1/DROSHA interaction and that APE1-kd is associated to increased oxidation levels of pri-miRNAs would support a major role of APE1 in the RNA-decay mechanisms of miRNA primary transcripts. Considering our previous data on APE1 endoribonuclease activity over abasic and oxidized RNA[13], the model described here could be generalized to the majority of oxidative stress-regulated miRNAs and possibly extended to all RNAs which undergo extensive damage including oxidation, alkylation and abasic lesions formation[19]. Interestingly, oxidative modification of mRNA seems to be highly selective, having an impact on the expression level of specific genes and on protein translation efficiency[24, 47]. Alterations of these pathways have a role in the pathogenesis of different human pathologies ranging from ageing to neurodegenerative and cancer diseases[22, 48, 49].

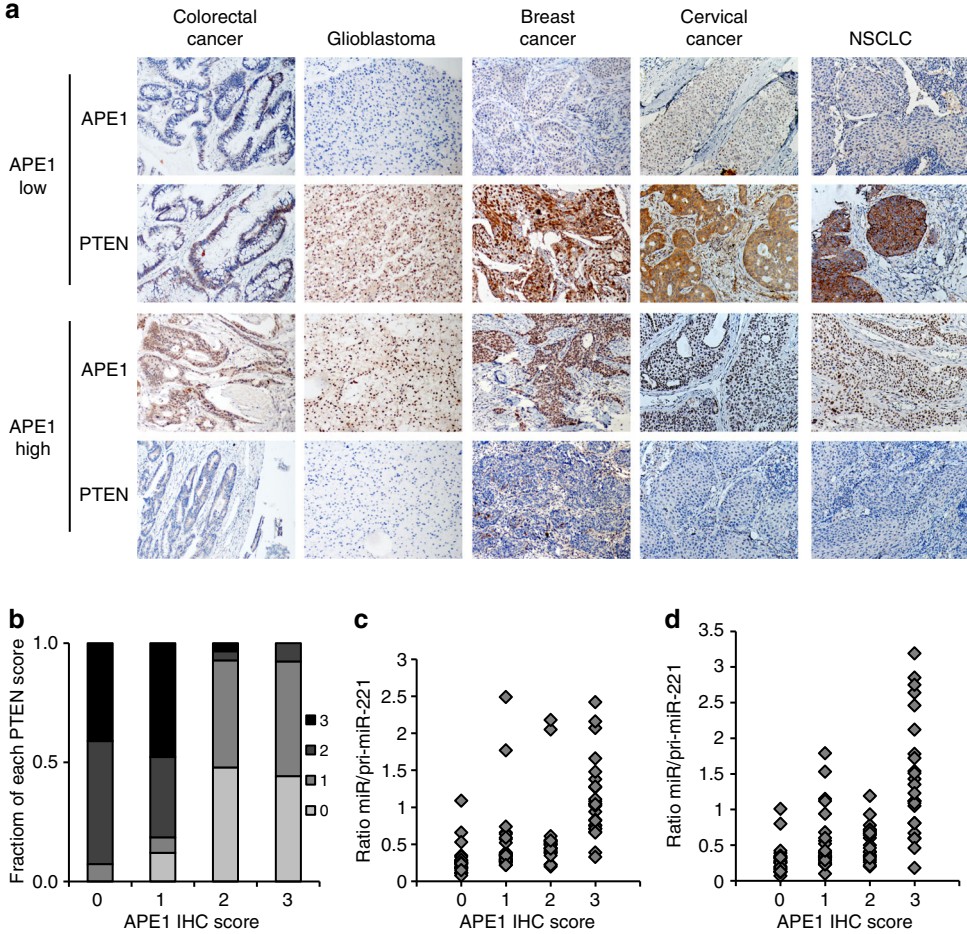

**Fig. 6** Correlative expression of APE1 and miR-221/222 with PTEN in a cohort of human cancer specimens. **a** APE1 and PTEN protein expression were determined by IHC assay and the representative images of both APE1 and PTEN were shown. PTEN expression significantly increased in tumor tissues showing poor APE1 expression, while was suppressed in tumor tissues showing high APE1 expression. **b** *Bar graph* showing the percentage of each score level of PTEN in 0, 1, 2, and 3 score level of APE1. Data were categorized as follow: (i) score 0, no expression in tumor cells; (ii) score 1, faint/barely perceptible partial expression in <10% of tumor cells; (iii) score 2, weak to moderate expression in >10% of tumor cells; (iv) score 3, strong expression in >10% of tumor cells. **c** and **d** Scattered plots showing distribution of miR to pri-miR ratios for miR-221 and miR-222 in each score level of APE1–protein staining, respectively

Wang et al. recently demonstrated that also miRNAs can be oxidatively modified by ROS, changing their binding properties from native targets to new ones, as in the case of miR-184[25]. It is worth noting that the duplex structure of pri-miRNAs precursor may represent the favorite target of APE1 activity[17]. Therefore, we should reinterpret the roles of APE1 in modulating cellular responses to genotoxic stresses and in the pathogenesis of human diseases, in light of the new role of this multifunctional protein in RNA biology.

miR-221 and miR-222 are two highly homologous miRNAs, tandemly encoded on the X-chromosome, whose overexpression has been recently described in several human malignancies, including thyroid papillary carcinomas, glioblastoma, prostate carcinoma, gastric carcinoma and others[50]. They both act as oncogenic miRNAs commonly targeting a cluster of genes with a key role in tumor inhibition, such as PTEN in tumor suppression, PUMA in apoptosis, TRPS1 in epithelial to mesenchymal transition and the cell cycle inhibitors p27$^{Kip1}$ and CDKN1C/p57 [50, 51]. Upregulation of miR-221/222 has been shown to confer radio-resistance, cell growth and invasion capabilities to different cancer cell types by suppressing the action of PTEN and their other critical target[29, 31, 52, 53]. Furthermore, upregulation of miR-221/222 has been associated with the development of multidrug

resistance and altered response to chemotherapy[54–56]. Therefore, restoring these anti-cancer genes expression by inhibiting miR-221/222 levels has been considered as a potential therapeutic strategy[50]. Interestingly, we noticed that the different pri-miR-221 and pri-miR-222 expression levels we measured, may be suggestive for independent expression by different promoters, as also confirmed by experimental data obtained from the FANTOM5 project[57]. In fact, in the 1174 human samples analyzed, the RLE-normalized promoter activity of miR-222 is on average 3.8-fold higher than that of miR-221. Therefore, in addition to a commonly thought polycistronic nature, miR-221 and miR-222 may be independently transcribed (Supplementary Fig. 9). Regarding the critical role of apoptosis-resistance and EMT in acquired resistance to radiotherapy, chemotherapy, and targeted therapy, our result that APE1-endonuclease activity can interfere with miR-221/222 biogenesis represents, to the best of our knowledge, a novel combinational therapeutic approach via using APE1 inhibitors to enhance efficacy of current cancer treatment.

PTEN is a well-known tumor suppressor gene that negatively regulates the major cell survival PI3K/AKT signaling pathway. Downregulation of PTEN, as a consequence of miR-221/222 overexpression, results in a constitutive activation of the PI3K/ AKT pathway, which in turn promotes cell transformation[58–60].

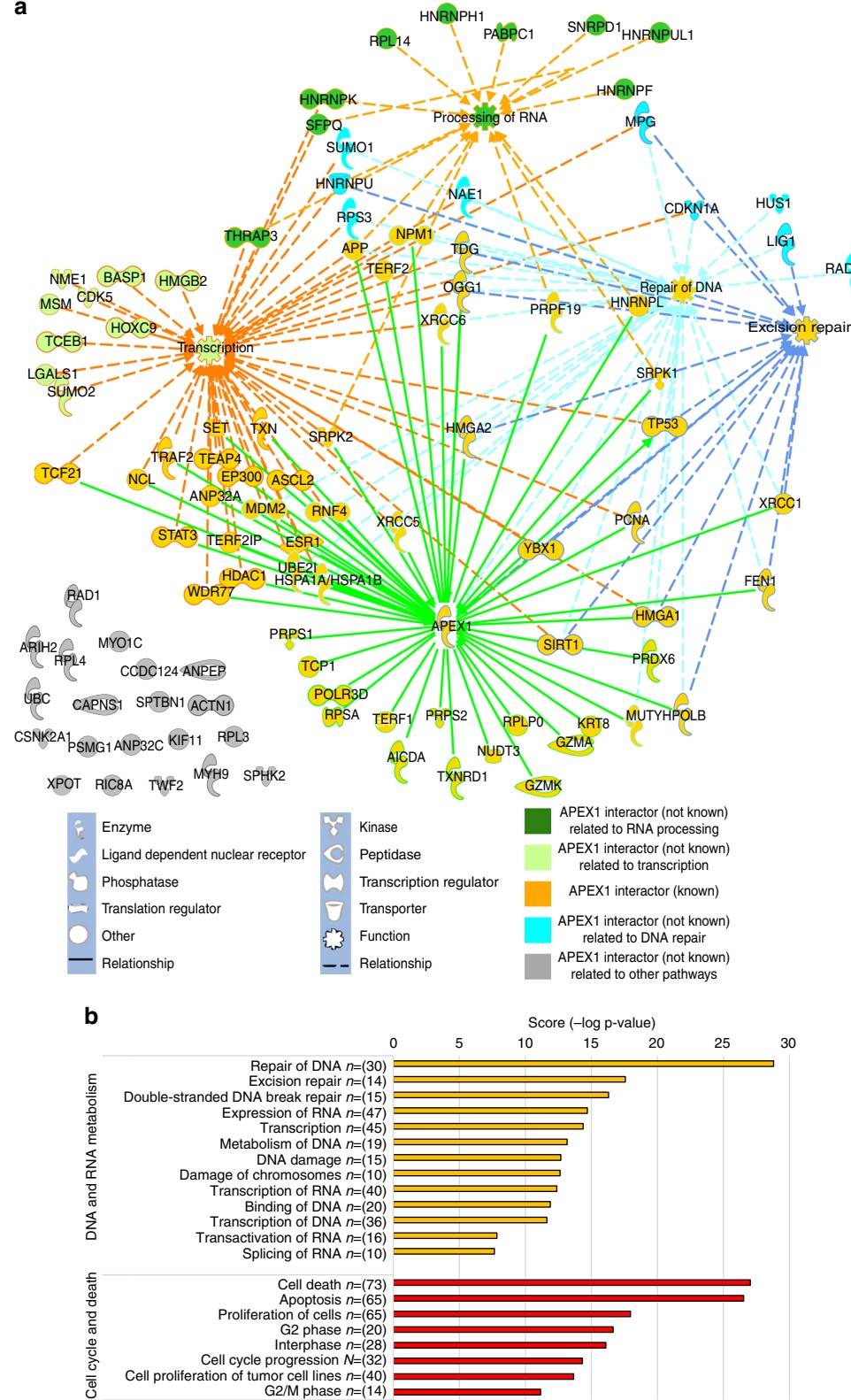

**Fig. 7** APE1–protein interactome network. **a** Protein network generated by Ingenuity Pathway Analysis. Starting from the gene list of the APE1-interacting protein partners, an interaction network was generated using the "connect tool" in order to differentiate between known or novel partners, also taking advantage of the Ingenuity Knowledge base. By using the "annotation tool", we selected and connected in the network the top four functional themes that were able to annotate the majority of the genes (63%). In the legend, we show the meaning of the shape representation and the molecular relationships in the IPA network. **b** *Bar chart* showing the enrichment of specific functions obtained using IPA for the APE1-binding protein partners. *Bars length* represent the –(log *P*-value) of the enrichment

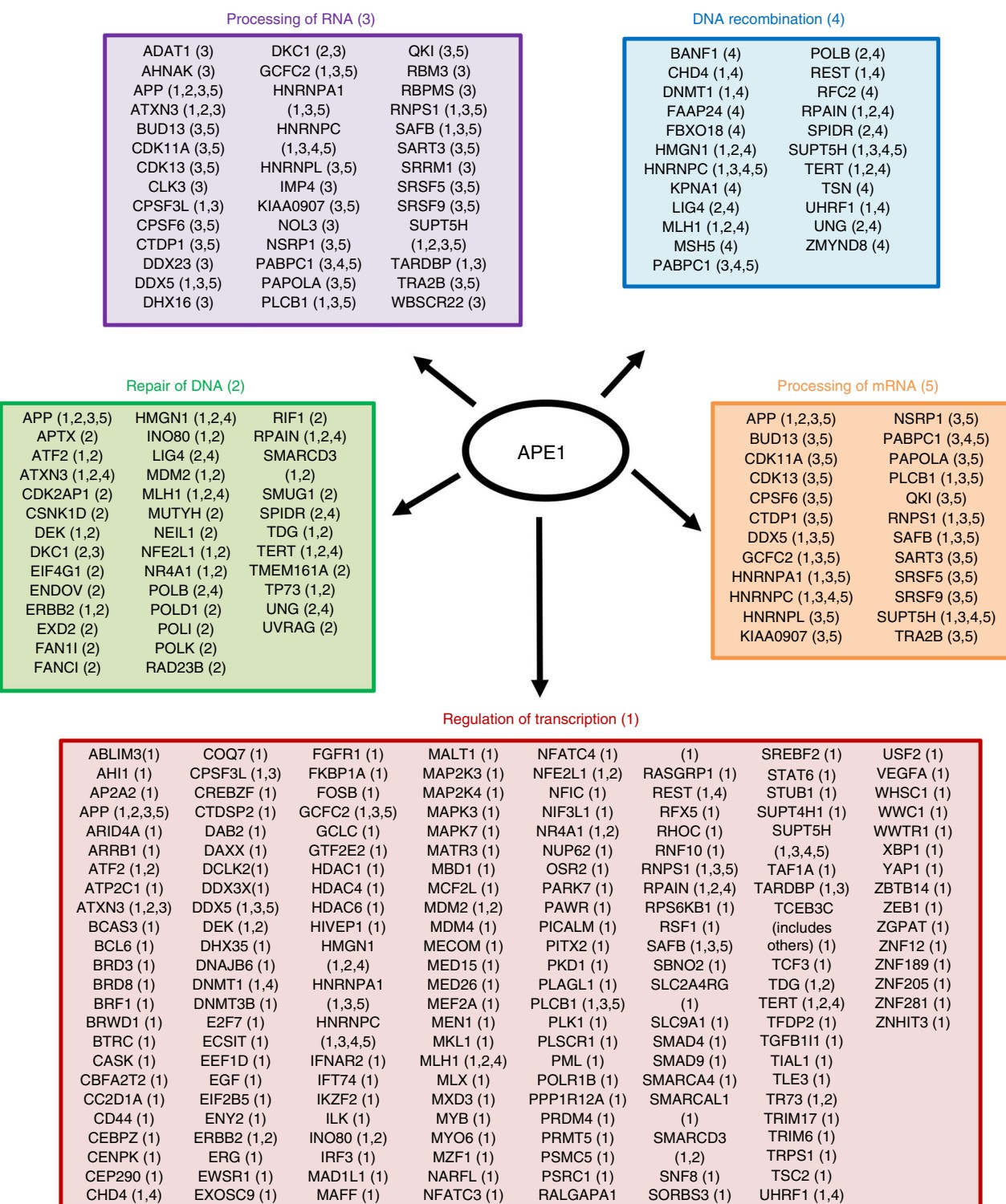

**Fig. 8** RIP identification of RNAs interacting with APE1. Top-five functional annotation clusters of APE1–RNA targets identified by Ingenuity Pathway Analysis based on functional terms of the "Biological Process" category. RNAs in more than one cluster are cross-referenced with numbers

These findings parallel our previous data[6], in which we observed that APE1 silencing, while blocking the Egr-1 mediated inducible expression of PTEN gene upon $H_2O_2$ or TSA treatments, was associated with a paradoxical upregulation of the PTEN protein itself under basal condition. Now, we propose a model in which APE1 acts as a central player in the homeostatic regulation of PTEN expression through transcriptional (via Egr-1) and

post-transcriptional (via miR-221/22) counteracting mechanisms involved in a feedback autoregulatory loop. This model could be generalized to a number of genes regulated by APE1, whose expression was previously considered a sort of paradox[3, 6]. These observations are also of particular interest for cancer development, in light of recent papers in which APE1 expression has been correlated with low-PTEN expression in high grade

**Table 2 The most enriched functional annotation disease clusters of APE1–RNA targets identified by Ingenuity Pathway Analysis**

|  | Diseases | *P*-value | Score | # Molecules |
|---|---|---|---|---|
| Cancer | Epithelial cancer | 1.11E−23 | 22.95 | 756 |
|  | Digestive system cancer | 6.91E−20 | 19.16 | 701 |
|  | Colon cancer | 3.46E−10 | 9.46 | 393 |
|  | Breast cancer | 1.15E−05 | 4.94 | 158 |
|  | Ovarian cancer | 2.78E−04 | 3.56 | 121 |
|  | Hematologic cancer | 1.16E−03 | 2.94 | 209 |
| Hematological | Anemia | 8.72E−06 | 5.06 | 39 |
|  | High risk myelodysplastic syndrome | 2.63E−04 | 3.58 | 3 |
|  | Lymphoproliferative disorder | 4.37E−04 | 3.36 | 203 |
| Neurological | Spinocerebellar ataxia | 1.92E−04 | 3.72 | 13 |
|  | Disorder of basal ganglia | 5.47E−04 | 3.26 | 64 |
|  | Early stage Alzheimer's disease | 6.37E−04 | 3.20 | 3 |
|  | Neural tube defect | 6.36E−05 | 4.20 | 19 |
|  | Movement disorders | 1.55E−04 | 3.81 | 89 |
|  | Degeneration of nervous system | 5.64E−04 | 3.25 | 29 |
| Behaviour | Autistic-like traits | 1.66E−03 | 2.78 | 2 |

*P*-value, enrichment score and number of molecules for each specific term are shown. *APE1* apurinic/apyrimidinic endonuclease 1

gliomas with poor prognosis[61]. Our confirmation, in a cohort of different tumors, that the expression of APE1 correlates with that of mature miR-221/222, and inversely with that of PTEN, reinforces the relevance of our hypothesis in human cancer. Interestingly, miR-221/222 were recently found to be post-transcriptionally dysregulated in AML patients[27], in which also the APE1-endonuclease function is impaired[16]. Thus, our data highlight an unexpected new mechanism through which APE1 overexpression may play a central role in chemoresistance through post-transcriptional mechanisms involving onco-miR-NAs regulation and onco-miRNAs decay; these findings may open new perspectives for cancer diagnosis and therapy.

Intimate cross-talks between miRNA-processing machineries and nuclear factors is an emerging field of study in the DDR pathways, which may reveal important aspects of regulation in cancer biology[62–64]. Under genotoxic stress conditions, acetylated p53 can control the transcription and processing of some pri-miRNAs through association with p68 (DDX5), an RNA helicase of the DROSHA microprocessor complex[65]. Since APE1 acts as a regulator and interacting partner of p53[66] and it is able to bind some pri-miRNAs[65], we may speculate that APE1 endoribonuclease activity is a part of the inducible mechanisms regulating the processing of specific pri-miRNAs, during DDR due to oxidative stress and alkylating treatment. In this context, APE1 should control the quality of the precursor pre-miRNAs in the nucleus during miRNA biogenesis. Several modulators have been reported to influence the processing of pri-miRNAs besides p68 and p53, such as p72/p82 (DDX17), ADAR1, hnRNP A1, KSRP, SMADs, BRCA, YAP, Lin28, mutp53, DDX1, ARS2, DR5, ERα, and ERβ[64]. Notably, a recent work demonstrated that an APE1-interacting protein, i.e., YB-1, regulates the biogenesis of miR-29b-2 by blocking the recruitment of the microprocessor complex and Dicer to its precursor, and upregulates the expression levels of the host transcripts of miR-221/222[67]. These results are suggestive that our novel findings on APE1, to the best of our knowledge, may delineate new fascinating perspectives in miRNA biology.

Remarkably, APE1 may have a general role in RNA processing, not limited to miRNAs regulation. Through RIP-seq analyses, more than 1000 transcripts were found to be bound by APE1; they have functional involvement in RNA processing, regulation of transcription, and DNA repair with profound relevance in cancer development. This result provides a definitive evidence on

APE1 participation in many pathways through post-transcriptional mechanisms[10–13]. By using microarray analyses on HeLa cells as a cancer cell model, we previously demonstrated that APE1 silencing is associated with profound changes in gene expression[14]. Comparison of these data with previous gene expression profiling of APE1-kd cells[14] highlighted that the majority of APE1-regulated genes are potential target of APE1-regulated miRNAs (Supplementary Data File 1) suggesting that a significant portion of APE1-target genes may be regulated through indirect mechanisms involving non-coding RNAs. This ability to regulate gene expression through non-coding RNAs will increase our understanding of the biological functions of APE1, thus providing novel basis for the use of APE1 as a druggable target for cancer therapy. Although additional molecular details need to be delineated, our discovery that APE1 is a novel player in miRNAs biogenesis opens a new scenario in the study of this fascinating multifunctional protein.

## Methods
**Cell lines and materials**. HeLa, HCT-116 (ATCC, Manassas, VA), and MCF-7 cell lines (Sigma-Aldrich, St. Louis, MO) were grown in Dulbecco's modified Eagle's medium (Invitrogen, Monza, Italy) supplemented with 10% fetal bovine serum (Euroclone, Milan, Italy), 100 U ml$^{-1}$ penicillin, 10 µg ml$^{-1}$ streptomycin sulphate. OCI/AML-2 and OCI/AML-3 cell lines (a kind gift by Emanuela Colombo) were grown in alpha-MEM (Euroclone) supplemented with 20% fetal bovine serum, 100 U ml$^{-1}$ penicillin and 10 µg ml$^{-1}$ streptomycin sulfate. HeLa cell clones expressing an ectopic APE1–FLAG-tagged form[13] were grown in Dulbecco's modified Eagle's medium supplemented with 10% fetal bovine serum, 100 U ml$^{-1}$ penicillin, 10 µg ml$^{-1}$ streptomycin sulfate, 3 µg ml$^{-1}$ blasticidin, 100 µg ml$^{-1}$ zeocine, and 400 µg ml$^{-1}$ geneticin (Invitrogen, Carlsbad, CA). For inducible APE1-shRNA experiments, doxycycline (1 µg ml$^{-1}$; Sigma-Aldrich) was added to the cell culture medium, and cells were grown for 10 days, as previously described[13, 14]. All cell lines were tested and free of mycoplasma contamination (N-GARDE Mycoplasma PCR Reagent, Euroclone).

**Transient transfections and cellular treatments**. One day before transfection, cells were seeded in 10-cm plates at a density of $3 \times 10^6$ cells per plate. Cells were then transiently transfected with 6 µg of plasmidic DNA using Lipofectamine 2000 Reagent (Invitrogen) according to the manufacturer's instructions and collected 24 h after transfection. APE1-kd HeLa cells were transiently transfected with expression plasmids for FLAG-tagged, siRNA-resistant APE1 mutants APE1$^{WT}$, APE1$^{N\Delta33}$, APE1$^{E96A}$, and APE1$^{C65S}$ which bears two mismatches in the cDNA sequence preventing the degradation of the ectopic APE1 mRNA, while leaving the APE1 amino acid sequence unaffected and with an empty-plasmid as a control.

For siRNA experiments, cell lines were transfected with 100 pmol siRNA APE1 5′-UACUCCAGUCGUACCAGACCU-3′ or the scramble control siRNA 5′-CCAUGAGGUCAUGGUCUGdTdT-3′ (Dharmacon, Lafayette, CO) using

DharmaFECT reagent (Dharmacon). After 72 h upon transfection, cells were collected and RNA extracted.

For APE1 endonuclease activity inhibition, HeLa cells were treated with 20 µM APE1 endonuclease inhibitor #3[33], 40 µM of fiduxosin[34], and 100 µM of E3330[35, 68] for the indicated time.

**NanoString nCounter system miRNA Assay.** miRNA expression profiling was performed with 100 ng of total RNAs form HeLa cell clones silenced for APE1–protein expression. RNA was isolated using the miRNeasy kit (Qiagen, USA) and samples were prepared for nCounter miRNA expression profiling using the human v2 miRNA expression panel, according to the manufacturer's recommendations (NanoString, Seattle, Washington, USA) in the Geneticlab Srl. Transcript counts were normalized through the normalization method incorporated in the model framework, estimating parameters from positive controls, negative controls, and housekeeping genes embedded in the nCounter system, using the NanoStringDiff package within Bioconductor[69]. Differential expression of genes was assessed on log2-normalized data with a generalized linear model likelihood ratio test, using the glm.LRT function within the NanoStringDiff package. A $q$-value cutoff of 0.1 was used to determine statistical significance. For clustering analysis, raw values were normalized using the NanoStringNorm package[70]. Starting from the log2-normalized values, genes with low standard deviation (SD < 0.2) were filtered out and hierarchical clustering of the samples was performed using Cluster 3.0 (http://bonsai.hgc.jp/~mdehoon/software/cluster/software.htm). The quality of the data was checked by the mean expression and SD values for housekeeping miRNAs and positive/negative controls (Supplementary Fig. 1c). Visualization of the clustering and of the heatmap of log2-normalized values were obtained using Java Treeview[71]. Independent validation analysis on 10 differential miRNAs was performed through qRT-PCR.

**Cumulative distribution function plot analysis.** The data set E-MEXP-1315[14], which was retrieved from Array-Express, was used to evaluate the differential gene expression between APE1-depleted and control cells. Standard procedures were used to obtain the log fold change for all the genes present in the microarray. Briefly, CEL files were loaded with Affy package, and Robust Multi-Array Average normalization was applied[72]. Statistical analysis for differentially expressed genes was performed with a linear model regression method using the Limma package[73]. $P$-values were adjusted for multiple testing using the Benjamini and Hochberg's method to control the false discovery rate[74]. Gene annotation was obtained from R-Bioconductor metadata packages, and the probesets were converted in Entrez Gene Id and Symbol Id, obtaining a differential mRNA expression matrix (DE-mRNA matrix). Starting from the differentially expressed miRNAs (Supplementary Data 1), we filtered out the features with $q > 0.01$ and absolute log fold change <1. For the remaining miRNAs ($n = 40$), we obtained the validated gene targets from the mirTarBase database[75]. Since, even with these constraints, the gene list was quite big ($n = 9326$), we decided to filter out genes that were not reported to be downregulated by at least two miRNAs, obtaining the final miRNA-targets gene list ($n = 5630$). Finally, we extracted from the DE-mRNA matrix the log fold change information corresponding to the obtained miRNA-targets gene list.

Then, we performed 1000 comparisons (using the Kolmogorov–Smirnov test and Wilcoxon test) in which the control vector was composed by the log fold change values randomly selected from the DE-mRNA matrix, while maintaining the size of log fold change of the miRNAs-targets gene list. The $P$-values were adjusted using the Benjamini–Hochberg method. Notably, the statistical tests were performed only on the one tail corresponding to the correct biological direction (increase of the miRNA-targets gene expression with respect to the control, $P = 6 \times 10^{-30}$ for KS test, and $P = 0.0016$ for Wilcoxon test). As a further control, we also checked in the opposite direction (decrease of the miRNA-targets gene expression with respect to the control), obtaining worst significant results ($P = 10^{-15}$ for KS test and $P = 1$ for Wilcoxon test). Finally, we choose a conservative approach to combine $P$-values averaging the log transformed $P$-values instead of using Fisher's method due to the dichotomous results ($P = 0$ for the correct biological direction tests and $P = 1$ for opposite direction). Empirical cumulative distribution function curves were calculated and plotted using the stats package inside the R/Bioconductor environment[76].

**RNA immunoprecipitation.** HeLa cell clones were seeded in 150-cm plates at a density of $1 \times 10^7$ cells per plate. Two 150-cm plates for APE1[WT]-expressing cells were grown. RIP[2, 42] was carried out as detailed in the Supplementary Information.

**Library preparation and sequencing.** TruSeq Stranded Total RNA with Ribo-Zero Human/Mouse/Rat (Illumina, San Diego, CA) was used for library preparation following the manufacturer's instructions. Both RNA samples and final libraries were quantified by using the Qubit 2.0 Fluorometer (Invitrogen) and quality tested by Agilent 2100 Bioanalyzer RNA Nano assay (Agilent technologies, Santa Clara, CA). Libraries were then processed with Illumina cBot for cluster generation on the flowcell, following the manufacturer's instructions and sequenced on 50 bp single-end mode with a HiSeq2500 apparatus (Illumina). The CASAVA 1.8.2 version of the Illumina pipeline was used to process raw data for both format conversion and de-multiplexing.

**Transcriptomics analysis.** Raw sequence files were subjected to quality control analysis using FastQC (http://www.bioinformatics.babraham.ac.uk/projects/fastqc/). The reads were quantified using Salmon[77, 78] to the human reference transcriptome GRCh38 obtained from the ENSEMBL web site. In particular, we utilized the combined FASTA of cDNA and ncRNA. Salmon performs transcript-level quantification estimates from RNA-seq data; it achieves its accuracy and speed via a number of different innovations, including the use of quasi-mapping (an accurate but fast-to-compute proxy for traditional read alignments) and a two-phase inference procedure that makes use of massively parallel stochastic collapsed variation inference[77, 78]. The obtained results were gene quantifications at the transcript level for each sample. Afterwards, we obtained differentially APE1-bound transcript lists using edgeR and applying the recommend procedures for the three comparisons (IP-APE1 vs. Input, IP-APE1 vs. IP-control, and, IP-control vs. Input)[79, 80]. In order to minimize the false positive elements, we selected as APE1-binding RNAs only the transcripts ($n = 1015$) that were detected as enriched in both the IP-APE1 vs. Input comparison (fold change >2, FDR < 0.05) and IP-APE1 vs. IP-control comparison (fold change FDR >2). Notably, none of these transcripts was enriched in the IP-control vs. Input.

**Functional analysis.** Data were analyzed through the use of David/EASE[46] and QIAGEN Ingenuity Pathway Analysis (QIAGEN Redwood City, www.qiagen.com/ingenuity). For IPA, transcripts were associated with biological functions/transcriptional regulators in the Ingenuity Knowledge Base. miRNA targets prediction was performed using the miRNA targets analysis included in IPA. A right-tailed Fisher's exact test was used to calculate a $P$-value to determine the probability that each biological function/transcriptional regulator assigned to the data set was due to chance alone.

**Cancer specimens and immunohistochemistry.** Ninety-four paraffin-embedded cancerous tissue samples, including NSCLC, colorectal cancer, breast cancer, cervical cancer, and glioblastoma, were collected from patients who underwent surgical resection without prior chemotherapy or radiotherapy in Daping Hospital, Third Military Medical University (Chongqing, China) from 2015 to 2016. This study was approved by the Ethics and Research Committee of the Daping faculty of Medicine, Third Military Medical University, Chongqing, China; written informed consents were obtained from all patients. The Histopathological assessment was carried out separately by two pathologists and then a consensus was made on discordant assessments. Sections from formalin-fixed and paraffin-embedded (FFPE) tumors were incubated with APE1 antibody (clone 13B8E5C2; dilution 1:5000; Novus Biologicals) or PTEN antibody (clone A2B1; dilution 1:100; Santa Cruz Biotechnology) overnight, at 4 °C. Sections were rinsed with phosphate-buffered saline (PBS) and incubated with goat anti-mouse secondary antibody. Sections were rinsed with PBS, developed with diaminobenzidine substrate, and then counterstained with diluted Harris hematoxylin. APE1 and PTEN staining were analyzed and scored for four categories: (i) score 0, no expression in tumor cells; (ii) score 1+, faint/barely perceptible partial expression in <10% of tumor cells; (iii) score 2+, weak to moderate expression in >10% of tumor cells; (iv) score 3+, strong expression in >10% of tumor cells. Image analysis was done by two experienced pathologists independently. Statistical significance was calculated according to the Spearman's rank correlation test.

**Quantitative real-time reverse transcriptase-PCR.** For the measurement of mRNA-expression from cell lines, total RNA was extracted with the SV Total RNA isolation System kit (Promega, Madison, WI). One microgram of total RNA was reverse transcribed using the iScriptcDNA synthesis kit (Bio-Rad, Hercules, CA), according to the manufacturer's instructions. qRT-PCR was performed with a CFX96 Real-Time System (Bio-Rad) using iQ SYBR Green Supermix (Bio-Rad). Primers used were: FOSB For 5′-AGCTAAATGCAGGAACCGG-3′, FOSB Rev 5′-ACCAGCACAAACTCCAGAC-3′; NEIL For 5′-GCCCTATGTTTCGTGGA-CATC-3′, NEIL Rev 5′-CGCTAGGTTTCGTAGCACATTC-3; POLB For 5′-AGTACACCATCCGTCCCTTG-3′, POLB Rev 5′-AAAGATGTCTTTTTCA CTACTCACTG-3′; PTEN For 5′-AAGTCCAGAGCCATTTCC-3′, PTEN Rev 5′-AATATAGGTCAAGTCTAAGTCG-3′; GAPDH For 5′-CCTTCATTGACCT-CAACTACATG-3′, GAPDH Rev 5′-TGGGATTTCCATTGATGACAAGC-3′. DNA was amplified in 96-well plates using the 2X iQ SYBR green supermix (Bio-Rad) and 10 µM of the specific sense and antisense primers in a final volume of 15 µl for each well. Each sample analysis was performed in triplicate. As negative control, a sample without template was used. The cycling parameters were denaturation at 95 °C for 10 s and annealing/extension at 60 °C for 30 s (repeated 40 times). In order to verify the specificity of the amplification, a melting-curve analysis was performed, immediately after the amplification protocol.

For pri-miRNA and mature miRNAs qRT-PCR analysis from in vitro cultured cell lines, RNA was isolated using miRNeasy kit (Qiagen, USA), according to the manufacturer's instructions. For pri-miRNA and mature miRNAs assay from FFPE samples, total RNA was extracted with the miRNeasy FFPE kit (Qiagen, USA). For pri-miRNA analysis, 1 µg of total RNA was reverse transcribed using the iScriptcDNA synthesis kit (Bio-Rad), according to the manufacturer's instructions. For mature miRNAs, TaqMan MicroRNA Reverse Transcription Kit (Applied Biosystems, Carlsbad, CA) was used. Briefly, 10 ng of total RNA were coincubated

with 1 µl of miRNA-specific primers (Assay-ID miR-221: 000524; miR-222: 002276; RNU44: 001094) and cDNA generated in the presence of RNase-inhibitors. Detection of successfully transcribed products was carried out using TaqMan2x Universal PCR Mix (no AmpErase UNG; Applied Biosystems) in combination with miRNA-specific primers and TaqMan-probes, and quantified on a Lightcycler 480 (Roche Applied Sciences). RT-qPCR results were calculated using the ΔΔct method, utilizing the expression of RNU44 and GAPDH as the housekeeping gene for mature miRNA and pri-miRNA, respectively.

**Isolation and quantification of abasic RNA.** Oxidatively depurinated/depyrimidinated RNA species were specifically labeled with an ARP, and isolated with streptavidin magnetic-beads based on the previous report[43]. In vitro synthesized RNA encoding *Xenopus* elongation factor 1α gene was added to the total RNA isolated from HeLa cells to 30 µg in total amount. The RNA was derivatized with 2 mM ARP in 50 mM Na acetate buffer pH 5.2 for 40 min, at 37 °C, and precipitated by ethanol. After resuspending with deionized water, an aliquot was put aside for quantifying total transcripts, and the rest of RNA was incubated with streptavidin magnetic-beads for 20 min, at 60 °C. After washing with wash buffers including urea, oxidized RNA was dissociated from beads by incubating with 2.5 mM biotin solution at 90 °C, for 5 min, and precipitated by ethanol. The precipitated oxidized RNA and total RNA were used for cDNA synthesis (Single strand cDNA synthesis kit with DNase, Thermo Fisher Scientific). Complementary DNA was employed for qRT-PCR with GTXpress Master mix and TaqMan probe for miR-221 or miR-222 (Hs03303007_pri or Hs03303011_pri, Thermo Fisher Scientific) using LightCycler 480, according to the manufacturer's protocol. The oxidation levels of miRNAs were determined based on ΔCq values between Cq of total RNA and oxidized RNA. Specificity of the qRT-PCR was confirmed by $T_m$ temperature and by testing without reverse transcription. PCR amplification was assessed by calibration curve using serial dilution method.

**Preparation of cell extracts and protein quantification.** For preparation of total cell lysates, cells were collected by trypsinization and centrifuged at $250 \times g$ for 5 min, at 4 °C. Supernatant was removed, and the pellet was washed once with ice-cold PBS and then centrifuged again as described before. Cell pellet was resuspended in lysis buffer containing 50 mM Tris-HCl (pH 7.4), 150 mM NaCl, 1 mM EDTA, and 1% w/v Triton X-100 supplemented with 1× protease inhibitor cocktail (Sigma), 0.5 mM phenylmethylsulfonyl fluoride (PMSF), 1 mM NaF and 1 mM Na3VO4 for 30 min, at 4 °C. After centrifugation at $12,000 \times g$ for 30 min, at 4 °C, the supernatant was collected as total cell lysate. The protein concentration was determined using Bio-Rad protein assay reagent (Bio-Rad, Hercules, CA). Blots were developed by using the NIR Fluorescence technology (LI-COR GmbH, Germany) or the ECL enhanced chemiluminescence procedure (GE Healthcare, Piscataway, NJ), the latter case indicated in each figure capture. Images were acquired and quantified by using an Odyssey CLx Infrared Imaging system (LI-COR GmbH, Germany) or by using a Chemidoc XRS video densitometer (Bio-Rad, Hercules, CA), respectively. Original uncropped images of western blots used in this study can be found in Supplementary Figs. 10 and 11.

**Antibodies and western blotting analysis.** For western blotting analyses, whole-cell lysates were prepared and 20 µg of proteins were resolved on 12% SDS–PAGE, transferred onto nitrocellulose membranes (Schleicher & Schuell Bioscience, Dassel, Germany) and probed with antibodies for PTEN (sc-7974; Santa Cruz Biotechnology, Santa Cruz, CA) (1:1000), APE1[14] (1:1000), FLAG (F1804, SIGMA) (1:5000), Akt1 (sc-1618, Santa Cruz Biotechnology) (1:1000) and p-Akt1/2/3 (Ser473)(sc-7985-R; Santa Cruz Biotechnology) (1:1000). Data normalization was performed by using monoclonal anti-actin and anti-tubulin (Sigma-Aldrich) as indicated. The corresponding secondary antibodies labeled with IR-Dye (anti-rabbit IgG IRDye 680 and anti-mouse IgG IRDye 800) were used. Detection and quantification was performed with the Odyssey CLx Infrared imaging system (LI-COR GmbH, Germany). The membranes were scanned in two different channels using an Odyssey IR imager; protein bands were quantified using Odyssey software (Image Studio 5.0) and the relative signal, expressed as ratio of the treated group over the control group, was calculated.

**Immunofluorescence confocal and proximity ligation analyses.** Immunofluorescence procedures and PLA were carried out as described earlier[41, 42]. To study the interaction between APE1 and DROSHA in vivo, we used the in situ Proximity Ligation Assay technology (Duolink, Sigma-Aldrich). After incubation with monoclonal anti-APE1 (1:22)[14] for 3 h, at 37 °C, cells were incubated with polyclonal anti-DROSHA (ab85027, Abcam, Cambridge, MA)(1:200) overnight, at 4 °C. PLA was performed following the manufacturer's instructions. Technical controls, represented by the omission of anti-DROSHA primary antibody, resulted in the complete loss of PLA signal. Cells were visualized through a Leica TCS SP laser-scanning confocal microscope (Leica Microsystems). Determination of PLA signal was performed using the Blob Finder software (Center for Image Analysis, Uppsala University).

**AP-site incision assays.** APE1 endonuclease activity was monitored using HeLa cell extracts as follows:[13, 34]. Briefly, enzymatic reactions were carried out in a

final volume of 10 µl using 12.5 ng of cell extracts in a buffer containing 50 mM Tris-HCl pH 7.5, 50 mM KCl, 10 mM MgCl₂, BSA (1 µg ml⁻¹), and 1 mM DTT. Extracts were incubated for 15 min, at 37 °C, with 100 nM of double-stranded 26-mer abasic DNA substrate containing a single tetrahydrofuranyl artificial AP site at position 14, which is cleaved to a 14-mer in the presence of AP endonuclease activity[34]. The double-strand DNA was obtained by annealing a 5′-DY-782-labeled oligonucleotide 5′-AATTCACCGGTACCFTCTAGAATTCG-3′ (where F indicates the tetrahydrofuran residue), with an unlabeled complementary sequence 5′-CGAATTCTAGAGGGTACCGGTGAATT-3′.

Reactions were halted by addition of formamide buffer (96% formamide, 10 mM EDTA and gel Loading Buffer 6× (Fermentas)), separated onto a 20% (w/v) denaturing polyacrylamide gel and analyzed on an Odyssey CLx scanner (Li-Cor Biosciences). The percentage of substrate converted to product was determined using the ImageStudio software (Li-Cor Biosciences).

**APE1-interactome analysis.** Immunopurified protein material from endogenous APE1-silenced HeLa cells expressing APE1^WT or APE1^NΔ33 and grown under different experimental conditions, or from endogenous APE1-silenced HeLa cells stably transfected with the empty vector and expressing a scrambled siRNA sequence (APE1^SCR) were analyzed in parallel by SDS–PAGE. After colloidal Coomassie staining, whole-gel lanes were cut into six slices, minced, and washed with water. Corresponding proteins were separately in-gel reduced, S-alkylated with iodoacetamide, and digested with trypsin, as detailed in Supplementary Information.

**Statistical analyses.** All reported values are represented as the mean ± SD of three biological replicates. Statistical analyses were performed by using the Student's t-test. $P < 0.05$ was considered as statistically significant. The use of other statistical tests has been indicated where required.

**Data availability.** Illumina reads are deposited into NCBI Sequence Read Archive under accession number SRP078114. The data that support the findings of this study are available from the corresponding author on request.

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

## Acknowledgements

We thank Dr. Gang Chen, Daxue Zhou, Yi Cheng, Yuxin Yang, and Lei Zhang for their technical support and Dr. Emanuele De Paoli for critically reading the Manuscript and advice in the NGS data analyses. We thank Dr. Emanuela Colombo for the AML-2 and AML-3 cell lines, and Dr. Kefei Yu for the APE1-null mouse B-cell line. We are grateful to Prof. Bruce Demple for helpful comments during preparation of the Manuscript. This paper is entitled in the memory of Prof. Franco Quadrifoglio a master of science and life and an unforgettable friend. This work was supported by the Associazione Italiana per la Ricerca sul Cancro (AIRC) [grant #: IG2013-14038 to G.T.]. M.L. was supported by a grant from the National Natural Scientific Foundation of China (grant #:81673029). Y.C. is supported by AIRC/FIRC 2015.

## Author contributions

G.T. designed and conceived the study and supervised the experiments; G.A. performed most of the experiments, analyzed the data, and critically contributed to the interpretation of the results; L.L., C.D.A., and A.S. carried out the proteomic analysis; M.T. performed the abasic RNA quantification studies; F.S., E.D., Y.C., S.P. performed the bioinformatics analysis of all the proteomics and genomics datasets; S.R. performed the NGS analysis; S.Z. and M.L. performed the histological analysis on tumor biopsies specimens. G.T. and G.A. mainly wrote the manuscript; S.P. and M.L. provided critical comments and suggestions and contributed to interpretation of the results and writing of the manuscript. All authors critically read and approved the final version of the manuscript.
