## [Peer Review File · Nature Communications]

Reviewers' Comments:

Reviewer #1 (Remarks to the Author)

The aim of this study was to investigate whether the APE1 endonuclease may indirectly regulate gene expression through post-transcriptional mechanisms involving miRNAs processing and/or RNA regulation. The authors characterized APE1 RNA- and protein-interactomes and identified a role for this BER enzyme in the processing of precursor miRNA. This occurs via an association of APE1 with the DROSHA complex occurring during oxidative stress conditions. More specifically they focused their attention on APE1-dependent processing of miR-221/222 and the regulation of PTEN expression. They also propose that APE1 participation in different RNA- and protein-interactomes involved in cancer development makes their findings relevant in the tumor biology of different type of cancers.

The manuscript contains new and interesting data indicating a novel role for APE1 beyond its function as a proper DNA repair enzyme. The authors used a series of appropriate and state-of-the-art technology to reach their conclusions.

These results might be of interest to a general audience. However some clarifications are needed before publication in Nature communications.

Major points

Page 10, lines 211-217 and Figure 2d and 2e: The authors claim" that there is a decrease in the expression of two mature miRNAs (miR-221/222) upon APE1-silencing". This is certainly true for miR-222 levels but does not seem to be significant for miR-221. The authors should modify the text and comment this difference.

Page 12 and Figure 4b: again also in this case the two miRNAs (miR-221/222) do not behave in similar ways (relative RNA levels are higher for pri-miR-221 and the ratio miR/pri-miR are increased for miR-222 and decreased for miR-221). These differences should be more clearly reported and discussed.

Minor points

Supplementary Fig. S1a should be Supplementary Fig. 1a.

HOS is the abbreviation for human osteosarcoma.

A brief description of scoring categories for APE1 and PTEN staining should be included in the legend to the Fig. 6 or in the text (presently only in M&M).

No reference to Supplementary Fig. 6b in the main text or in the legend to the Figure. A full explanation of this Table (?) is required including what is the significance f number in parenthesis.

Reviewer #2 (Remarks to the Author)

The study here has explored the hypothesis that APE1 plays general and specialized roles in RNA metabolism, which in some instances can affect gene expression. The authors provide some of the first evidence that APE1 can process specific miRNA species (i.e. miR221/miR222), and in doing so, can seemingly affect chemoresistance through regulation of PTEN status. Additional studies indicate that APE1 interacts with the DROSHA machinery, in a manner that is influenced by stress conditions and RNA damage, and associates with numerous proteins involved in RNA metabolism and several RNA species. Though many of the presented concepts have been described in some

form or another previously, the manuscript here looks at the issue using a set of broader omics approaches (experimental and computational), while focusing on miR221 as a representative example of APE1-mediated RNA regulation. While the studies offer new insights into APE1 functions, particularly with regards to RNA metabolism, there are a number of disconnects between the different elements of the paper, as well as some significant weaknesses that require further development.

1. I don't see the relationship between the miRNome of hydrogen peroxide treated cells and APE1-kd cells. Consistently, the similarities between these two biological entities is quite limited. Certainly, hydrogen peroxide could have numerous effects on gene expression responses that don't relate to APE1 functions. As such, I feel the hydrogen peroxide experiments and results should be omitted from the paper.
2. There appears to be an increase in the miR/pri-miR ratio in the E3330 treated cells. Is that significant?
3. The E3330 extract should be added to the in vitro cleavage assays of Fig 3A, to determine the inhibitor's effect on APE1 incision activity. Also in Fig 3A, why does the FDX inhibitor reduce APE1 incision activity? While this inhibitor may affect APE1 localization (as suggested in the Results section), since total cell extracts are being assayed, the result doesn't make obvious sense. Please explain.
4. The description of the AML cell experiment is poor (Fig 3B). What is the difference between the AML2 and AML3 cell line? I assume there is a difference in APE1 localization, but which has higher cytosolic localization is not clear. Also, if there is increased APE1 distribution to the cytoplasm, wouldn't that cause an increase in the miR/pri-miR ratio (assuming processing takes place in the cytoplasm and not the nucleus)? This issue should be addressed once the cell lines have been better described.
5. A cleaner and necessary complementary experiment to using the different inhibitors is to complement the KD cells with various site-specific mutants of APE1 that (i) inactivate its nuclease activities; (ii) disrupt its interaction with NPM; and (iii) prevent its redox activity. The authors have a good deal of experience with such approaches and possess the necessary reagents, and this analysis would address concerns about potential off-target effects of the inhibitor molecules. It would also allow for more sound mechanistic conclusions.
6. While the data are more or less convincing, a nice (additional) control experiment would be to run the PLA (Fig 4A) in the HeLa APE1-kd cells after hydrogen peroxide treatment. Presumably in this situation, no signal would be observed at any time point.
7. The authors seem to ignore the fact that their rescue experiment does not restore pAKT1 levels back to normal (Fig 5C). Any explanation?
8. Although there does appear to be a relevant correlation between APE1, miR-221 and PTEN "expression" in their cancer patient studies, there are also several outliers (Fig 6, see for example panels C and D). This is consistent with biology being complex, but indicates that there are multiple mechanisms at play. I might be more cautious in interpreting these results with respect to their APE1-miR221-PTEN model. Indeed, a paper focused more on the mechanistic aspects of the work (see next two comments), leaving out the patient analysis, might have greater appeal.
9. It could be made clearer in the protein-interactome studies (p. 15-16) which results were from published reports and which are from the findings conducted here. Moreover, having all of the findings in Supplementary Material makes it difficult to evaluate the actual experimental work. In many ways, the interactome studies (Fig 7, 8 and the collection of Sup Figs) would be better served if more comprehensively presented and described.
10. It's not clear how the APE1 immunoprecipitation studies were conducted in the interactome analysis. Specifically, were whole cell extracts used? And if so, was there any effort to distinguish between cytoplasmic and nuclear APE1 complexes? I assume much of the miRNA processing takes place in the cytoplasm, but perhaps that can be clarified. Also, it's unclear why RNase treatment might affect the interactions of APE1 with POLB and XRCC1 (Sup Fig 6A). Does this imply a broader role for DNA repair proteins in RNA metabolism?
11. The English grammar, while generally understandable, is poor throughout.

Reviewer #3 (Remarks to the Author)

APE1 is a well-studied protein important in DNA damage and repair, and acts as a regulator of many cancer-related genes. In this manuscript, the authors unravel a previously uncharacterized function of APE1 in miRNA processing. They propose that APE1 interacts with Drosha to influence pri-miRNA processing and modulate the maturation of miR-221 and miR-222, two known oncogenic miRNAs that act via the PTEN pathway. The APE1-Drosha interaction is enhanced upon genotoxic stress to promote miR-221/-222 processing and resulting in down-regulation of the tumour suppressor PTEN. The authors also further analysed APE1 protein- and RNA-interactome. Overall, the findings presented here may provide insights into cancer biology and information relevant to therapeutics. However, the major conclusion that APE1 plays significant roles in gene regulation via the regulation of miRNA processing should be supported by stronger evidence. With such supporting results (Potential experiments are suggested in the major points section below.), this manuscript could be a good candidate for publication in Nature Communications.

Major points:

1. Page 13: The hypothesis "The effect of APE1 on the PTEN-pathway is dependent on miR-221/222" (Section heading) is not tested in this study. To support this conclusion, the authors should inhibit mir-221/222 activity and demonstrate that APE1 knockdown less strongly affects the PTEN level when the miRNA activity is inhibited. Alternatively, PTEN 3'UTR sensor assays could be performed and test 1) whether the up-regulation of PTEN in APE1-KD cells is mediated by a post-transcriptional mechanism through the 3'UTR and 2) whether the up-regulation of the PTEN 3'UTR sensor is dependent on the miR-221/-222 target sites by introducing point mutations. In the discussion section, the authors noted that "the majority of APE1-regulated genes are potential targets of APE1-regulated miRNAs" (Page 21, line 477). This would be a strong result demonstrating the importance of miRNA-mediated gene regulation by APE1, and the data should be shown. Cumulative distribution function plot using expression fold-change values for targets and non-targets of APE1-regulated miRNAs would be sufficient.
2. Although the PLA results show a stress-enhanced interaction between Drosha and APE1, the lack of biochemical evidence is worrisome, especially when Drosha does not seem to appear as an APE1 partner in the proteomics data. Biochemical experiments to support the interaction (e.g. co-IP assays by detecting proteins by Western blotting) perhaps from the cells treated with H2O2 should be performed.
3. I'm puzzled why miR-222 was more strongly down-regulated in APE1 knockdown cells (Figure 2d), while only miR-221 was identified as a differentially expressed miRNA in cells stressed by H2O2 (Figure 1b). Figure 4b is even more confusing, why are mir-221 and mir-222 showing distinct behaviors? Any explanation?

Minor points:

1. "Precursor miRNA" and "pre-miRNA" are generally used to refer to the short hairpin that is released by Drosha, not the primary transcripts. In this manuscript, the word "precursor" is used when referring to primary transcripts, which is somewhat confusing. The author should avoid this.
2. It is unclear the figure for in vitro cleavage assay is part of Fig 3a or 3b.
3. In Figure 3b, it is unclear what OCI/AML-2 and OCI/AML-3 cell lines are and the difference between the two.
4. Line 305 refers to abbreviated NSCLC. The full description of NSCLC can only be found in the Methods section.
5. Figures should be labeled more accurately. Figure 2C Y-axis has duplicated numbers, Figure 5C Y-axis should be "relative protein level" not "fold change", and Figure 6B Y-axis would not be "%" (I am assuming it shows the fraction of each category.), just to point out a few. In the main text, when the authors discuss the correlation (page 14), some values are shown as "r" and others as "r²". The statistical method used should be indicated and a consistent method should be used. Line 318, I assume that "r²=418" is incorrect.
6. Throughout the manuscript, there are sentences that I could not comprehend. Just to point out

one, I could not understand what this sentence means: Page 22 line 479-482 "Whether its ability to... its importance as a druggable target for cancer therapy will be definitely demonstrated.", for example. Also, besides the issue with the sentence structure, the last part of this sentence regarding the importance of the finding in therapeutics may be too strong, because the present study still does not directly clarify the extent to which APE1 regulates genes via regulatory non-coding RNAs, particularly in cancers.

Point-by-point answers to Referees' requests

Reviewer #1:

We thank this Reviewer for his/her very encouraging comments and constructive remarks (stating that: "*The manuscript contains new and interesting data indicating a novel role for APE1 beyond its function as a proper DNA repair enzyme. The authors used a series of appropriate and state-of-the-art technology to reach their conclusions. These results might be of interest to a general audience*"). These comments helped us prepare an improved version of the Manuscript. In our opinion, we have addressed all suggestions, and a list of the answers to the specific points raised is following:

Major points

Q1.1. Page 10, lines 211-217 and Figure 2d and 2e: The authors claim" that there is a decrease in the expression of two mature miRNAs (miR-221/222) upon APE1-silencing". This is certainly true for miR-222 levels but does not seem to be significant for miR-221. The authors should modify the text and comment this difference.

A1.1. We thank this Referee for the point raised. The levels of the two mature miRNAs (Fig. 2d) were measured before and after APE1-silencing. miR-222 expression resulted about 6-fold more abundant than that of miR-221 in HeLa control cells (SCR). miR222 was significantly decreased upon APE1 silencing (siRNA). In the case of miR-221, we observed a certain trend of decrease, although not statistically significant. On the other hand, the amount of each pri-miR was increased upon APE1 silencing, as confirmed from the corresponding miR to pri-miR ratios (Fig. 2e). In order to substantiate the essential role of APE1 for the maturation process of miR-221/222, we used a recently developed APE1-ko mouse cell model (i.e. CH12F3) (Masani et al., Mol Cell. Biol., 2013). Data obtained in this case showed a direct and clear relationship between the expression of APE1 and the expression levels of mature miR-221/222 (Supplementary Fig. 2). Overall, these results indicate that, although to a different extent possibly dependent on the absolute miRNAs expression levels, or to a different kinetics in the turnover ratio of each miRNAs, the pri-miR-221/222 processing is compromised in APE1-depleted cells. Therefore, we modified the text accordingly (Page 10, Lines 222-229).

Q1.2. Page 12 and Figure 4b: again also in this case the two miRNAs (miR-221/222) do not behave in similar ways (relative RNA levels are higher for pri-miR-221 and the ratio miR/pri-miR are increased for miR-222 and decreased for miR-221). These differences should be more clearly reported and discussed.

A1.2. We thank this Referee for this comment. We clarified this point as follows. This experiment was performed in HeLa cells and not in APE1-kd cell clones; all RNA levels were expressed as relative to untreated controls (NT) and not as absolute values, as in Figure 2c. This point has been better clarified in the amended legend of the vertical axis of each panel. Thus, the different kinetics observed in the case of the two miRNAs, particularly once starting with the release time upon H₂O₂-treatment (indicated as time 0 of release), may be ascribed to a different turnover rate of the two miRNAs. These comments have been now added in the text (Page 14, Lines 313-315).

Minor points

Q1.3. *Supplementary Fig. S1a should be Supplementary Fig. 1a.*

A1.3. We thank the Reviewer for this comment; the text was corrected accordingly.

Q1.4. *HOS is the abbreviation for human osteosarcoma.*

A1.4. This abbreviations is indeed described in the Results section (Page 8, line 171)

Q1.5. *A brief description of scoring categories for APE1 and PTEN staining should be included in the legend to the Fig. 6 or in the text (presently only in M&M).*

A1.5. According to the present request, we have now introduced a short description of the scoring categories in the legend to Fig. 6.

Q1.6. *No reference to Supplementary Fig. 6b in the main text or in the legend to the Figure. A full explanation of this Table (?) is required including what is the significance of number in parenthesis.*

A1.6. We thank the Reviewer for this comment. We have now introduced the reference for Supplementary Fig. 6b, together with its legend in the text (Page 17, line 385).

Reviewer #2:

We thank this Reviewer for his/her very encouraging comments and constructive remarks that helped us to prepare an improved version of the Manuscript, which takes into account the criticisms raised. We have addressed all suggestions and a list of answers to the specific points raised is following:

Q2.1. I don't see the relationship between the miRNome of hydrogen peroxide treated cells and APE1-kd cells. Consistently, the similarities between these two biological entities is quite limited. Certainly, hydrogen peroxide could have numerous effects on gene expression responses that don't relate to APE1 functions. As such, I feel the hydrogen peroxide experiments and results should be omitted from the paper.

A2.1. As described in the text, the rationale for the experiments on the miRNome after H₂O₂-treatment was to identify those miRNAs regulated by genotoxic treatment in HeLa cells and that could be involved in the regulation of PTEN, a known APE1 target gene involved in chemoresistance. Starting from this hypothesis, we identified miR-221/222, which were further investigated. As stated in the text, we did not combine both conditions (i.e. oxidative stress and APE1-kd) in order to avoid selection of off-target effects due to non-specific triggering of DNA damage response (DDR) by simultaneously exposing the cells to oxidative damage in a context of BER-deficiency. Having said that, we believe that removing this part from the paper could generate some confusion in the Readers.

Q2.2. There appears to be an increase in the miR/pri-miR ratio in the E3330 treated cells. Is that significant?

A2.2. The slight increase of the miR-222/pri-miR ratio we observed upon E3330 treatment is statistically significant and it is coherent with the same extent of increase obtained in the new experiment we performed once expressing the redox-defective mutant C65S (novel Figure 3b). We actually do not have any explanation for this increase; it may be associated with secondary phenomena related to the inhibition of the redox function of APE1 or to the expression of a redox-defective mutant protein that causes as a main biological effect a mitochondrial impairment (Vascotto et al., Mol Cell Biol 2011). Further experiments are required to elaborate a realistic rationale for this observation. A comment has been added in the text to highlight this observation (Pages 11-12, Lines 246; 251-262).

Q2.3. The E3330 extract should be added to the in vitro cleavage assays of Fig 3A, to determine the inhibitor's effect on APE1 incision activity. Also in Fig 3A, why does the FDX inhibitor reduce APE1 incision activity? While this inhibitor may affect APE1 localization (as suggested in the Results section), since total cell extracts are being assayed, the result doesn't make obvious sense. Please explain.

A2.3. According to Reviewer suggestion, we added the *in vitro* cleavage assays with cell extracts from E3330-treated cells. Data obtained (Fig. 3a and Supplementary Fig. 3) clearly show that the treatment with the redox inhibitor did not exert any apparent effect on the enzymatic activity of APE1 protein.

FDX inhibitor treatment displaces the interaction between APE1 and NPM1 (Poletto et al., Mol Carcinogenesis 2015). We already proved that this interaction positively regulates the endonuclease activity of the protein within cells (Fantini et al., Nucleic Acid Res., 2010 and Vascotto et al., Oncogene 2014). Moreover, we previously demonstrated that this inhibitor

significantly impairs the endonuclease activity of the protein, in a way comparable to that of compound #3 (see Fig. 6c in Poletto et al., Mol. Carcinogenesis 2015). Therefore, the rationale for the use of this inhibitor is due to its inhibitory function on the enzymatic activity targeting the interaction of APE1 with NPM1 within cells. These aspects have been now clarified also in the amended text (Page 11, lines 238-240).

Q2.4. The description of the AML cell experiment is poor (Fig 3B). What is the difference between the AML2 and AML3 cell line? I assume there is a difference in APE1 localization, but which has higher cytosolic localization is not clear. Also, if there is increased APE1 distribution to the cytoplasm, wouldn't that cause an increase in the miR/pri-miR ratio (assuming processing takes place in the cytoplasm and not the nucleus)? This issue should be addressed once the cell lines have been better described.

A2.4. We have better described the difference between the AML cell clones in the Results section (Page 12, Lines 263-264) and in the Figure Legend to Fig. 3c. In particular, OCI/AML3 cells, which stably express a NPM1c+ mutant protein relocalizing to the cytoplasmic compartment, have a significant accumulation of cytoplasmic APE1 as a consequence of its interaction with the aberrant protein NPM1 and a consequent BER-impairment, as we previously demonstrated¹⁶. Conversely, the OCI/AML2 line represents the control cell clone expressing a wild-type NPM1 protein, which correctly accumulates within nucleoplasm and nucleoli.

We believe that the effect of APE1 in modulating miR-processing takes place in the nucleus since we have observed the accumulation of pri-miRNA species (*i.e.* molecules that are exclusively processed in the nucleus) upon APE1 silencing (Figure 2). Thus, the impairment observed in OCI/AML3 cells, which present APE1 mislocalization in the cytoplasm and loss of BER-activity in the nucleus (Vascotto et al., Oncogene, 2015), is in perfect agreement with this hypothesis.

Q2.5. A cleaner and necessary complementary experiment to using the different inhibitors is to complement the KD cells with various site-specific mutants of APE1 that (i) inactive its nuclease activities; (ii) disrupt its interaction with NPM; and (iii) prevent its redox activity. The authors have a good deal of experience with such approaches and possess the necessary reagents, and this analysis would address concerns about potential off-target effects of the inhibitor molecules. It would also allow for more sound mechanistic conclusions.

A2.5. According to Reviewer suggestion, we performed additional experiments with three APE1-defective mutants: i) the E96A that reduces the enzymatic activity of the protein (Barzilay G et al., Nucleic Acid Res., 1995); ii) the NΔ33 deletion mutant that loses the interaction with NPM1 (Vascotto et al., 2009; Fantini et al., Nucleic Acid Res., 2010); the C65S that abolishes the redox activity of APE1 (Walker LJ et al., Mol Cell Biol. 1993). Resulting data perfectly paralleled the results obtained with the specific inhibitors; they are reported in the novel Fig. 3b. An additional experiment was also performed on APE1-knock out cells, which were recently developed by Masani et al, (2013), further confirming the requirement of APE1 for functional expression of mature miR-221/222 (Supplementary Fig. 2a).

Q2.6. While the data are more or less convincing, a nice (additional) control experiment would be to run the PLA (Fig 4A) in the HeLa APE1-kd cells after hydrogen peroxide treatment. Presumably, in this situation, no signal would be observed at any time point.

A2.6. According to Reviewer suggestion, we performed this control experiment. As expected (Supplementary Fig 4c,d), APE1 kd resulted in a significant reduction in PLA signals upon H₂O₂-treatment, paralleling the reduced interaction between APE1 and Drosha in APE1-kd cells.

Q2.7. *The authors seem to ignore the fact that their rescue experiment does not restore pAKT1 levels back to normal (Fig 5C). Any explanation?*

A2.7. We agree with the Reviewer comment on the partial rescue of pAKT levels upon APE1 re-expression. This may be due to the complex mechanisms responsible for the regulation of the phosphorylation status of this kinase, which is known to depend not uniquely on the activity of PTEN phosphatase (Di Cristofano, *Curr Top Dev Biol*, 2017 and Rodgers et al., *Biosci Rep*. 2017). In the amended text, we have now referred to this partial rescuing effect (Page 15, Lines 337-339).

Q2.8. *Although there does appear to be a relevant correlation between APE1, miR-221 and PTEN “expression” in their cancer patient studies, there are also several outliers (Fig 6, see for example panels C and D). This is consistent with biology being complex, but indicates that there are multiple mechanisms at play. I might be more cautious in interpreting these results with respect to their APE1-miR221-PTEN model. Indeed, a paper focused more on the mechanistic aspects of the work (see next two comments), leaving out the patient analysis, might have greater appeal.*

A2.8. PTEN is a well-known oncosuppressor gene, which is often down-regulated in cancerous tissues compared to paracancerous tissues. APE1 is often elevated in non-small cell lung cancer tissues, according to the results we obtained using immunohistochemistry assays. miR-221/222 is also considered to be a oncomiR in several types of human tumors. While APE1, PTEN and miR-221/222 are all associated with cancer, it would be interesting to investigate on the correlation between these molecules, especially whenever the regulatory mechanisms are also effective in *in vitro* systems.

Regarding the outliers in panels C and D of Figure 6, as the Reviewer already mentioned, it is consistent with biological complexity, and additionally, it might be partially due to the heterogeneity of tumor itself and might vary among different types of cancer. Although there are some exceptions in the results from clinical samples in our study, it is common that results from clinical samples are not always as clean as those from *in vitro* experiments. We believe that these results strengthened our findings described in the manuscript by confirming the regulatory relationship between APE1, miR-221 and PTEN *in vivo*, and could be meaningful for clinicians considering its general clinical significance.

Q2.9. *It could be made clearer in the protein-interactome studies (p. 15-16) which results were from published reports and which are from the findings conducted here. Moreover, having all of the findings in Supplementary Material makes it difficult to evaluate the actual experimental work. In many ways, the interactome studies (Fig 7, 8 and the collection of Sup Figs) would be better served if more comprehensively presented and described.*

A2.9. For the sake of clarity and according to Reviewer suggestion, we have now highlighted, in the list of APE1 interacting partners used for bioinformatics analysis (Supplementary Table 6), those related to the results we are publishing in this manuscript. The remaining proteins are related to previous literature data.

While we fully agree with the Reviewer on the opportunity to include (in the Main Manuscript) all the data regarding the interactome studies, we are forced by space constraints of the Journal to leave this part in the Supplementary Material.

Q2.10. *It's not clear how the APE1 immunoprecipitation studies were conducted in the interactome analysis. Specifically, were whole cell extracts used? And if so, was there any effort to distinguish between cytoplasmic and nuclear APE1 complexes? I assume much of the miRNA processing*

takes place in the cytoplasm, but perhaps that can be clarified. Also, it's unclear why RNase treatment might affect the interactions of APE1 with POLB and XRCC1 (Sup Fig 6A). Does this imply a broader role for DNA repair proteins in RNA metabolism?

A2.10. All the interactomic analyses were performed on total cell extracts with the aim to prevent possible artifacts due to subcellular fractionation procedures and to have a representative interactomic network considering the relative abundance of the different protein species. This has been now clearly stated in the text (Page 17, Lines 372-374). Indeed, the interaction of APE1 with DROSHA and its processing activity on pri-miR would suggest a role of APE1 on nuclear compartment. This aspect has now been clarified. The effect of RNase-treatment on interaction of APE1 with POLB and XRCC1 are suggestive for a role of RNA molecules in mediating interaction with other BER components and for a possible role of these enzymes in RNA processing/decay events. Experiments are actually ongoing in our laboratories to clarify this unexpected findings as well as experiments to map the APE1-protein interactome network in subfractionated cell extracts.

Q2.11. *The English grammar, while generally understandable, is poor throughout.*

A2.11. The manuscript has now been read and corrected by a native speaker.

Reviewer #3

We thank this Reviewer for his/her very encouraging comments and constructive remarks (stating that: “Overall, the findings presented here may provide insights into cancer biology and information relevant to therapeutics” and that “...this manuscript could be a good candidate for publication in *Nature Communications*”). They prompted us to prepare an improved version of the Manuscript facing the highlighted weaknesses. We carefully considered all suggestions and a list of answers to the specific points raised does follow:

Major points

Q3.1. Page 13: The hypothesis “The effect of APE1 on the PTEN-pathway is dependent on miR-221/222” (Section heading) is not tested in this study. To support this conclusion, the authors should inhibit mir-221/222 activity and demonstrate that APE1 knockdown less strongly affects the PTEN level when the miRNA activity is inhibited. Alternatively, PTEN 3’UTR sensor assays could be performed and test 1) whether the up-regulation of PTEN in APE1-KD cells is mediated by a post-transcriptional mechanism through the 3’UTR and 2) whether the up-regulation of the PTEN 3’UTR sensor is dependent on the miR-221/-222 target sites by introducing point mutations. In the discussion section, the authors noted that “the majority of APE1-regulated genes are potential targets of APE1-regulated miRNAs” (Page 21, line 477). This would be a strong result demonstrating the importance of miRNA-mediated gene regulation by APE1, and the data should be shown. Cumulative distribution function plot using expression fold-change values for targets and non-targets of APE1-regulated miRNAs would be sufficient.

A3.1. We agree with this Reviewer comment on the need of additional experiments to definitively prove that the APE1-dependent PTEN-pathway is directly dependent on miR-221/222. These experiments are in program in our Lab. However, they require a large amount of additional work and reagents currently unavailable in our Lab. Since the main aims of this work were to demonstrate a general role for APE1 in miRNA processing and in light of novel experiments already added in this revised manuscript (see below), and taking care of the space restrictions of the Journal, we prefer to keep these experiments for a further work. To avoid the presence of misleading sentences in this study, we have changed the Section heading accordingly, as follows: “The effect of APE1 inhibition on the PTEN-pathway correlates with miR-221/222 expression”.

In addition, following the Reviewer suggestion, we added an analysis in which we compared the cumulative distribution of expression changes for gene that were miRNA targets versus those of random mRNAs in the same cellular context. Notably, we were able to successfully integrate the data from the gene expression dataset produced several years ago (Vascotto et al., *Proteomics*, 2009) with the miRNA expression data reported in this manuscript. Resulting data have now been shown in Figure 1c, and are also explained in the main text (Pages 8-9, Lines 184-192). We also added the corresponding procedure in the experimental section. Finally, we added a list (Supplementary Table 1) showing the most significant miRNA found dysregulated in APE1-kd cells and their target genes dysregulated in APE1-kd cells (Vascotto et al., *Proteomics*, 2009).

Q3.2. Although the PLA results show a stress-enhanced interaction between Droscha and APE1, the lack of biochemical evidence is worrisome, especially when Droscha does not seem to appear as an APE1 partner in the proteomics data. Biochemical experiments to support the interaction (e.g. co-IP assays by detecting proteins by Western blotting) perhaps from the cells treated with H₂O₂ should be performed.

A3.2. The nature of the interaction between APE1 and Drosha should be weak and transitory, as it would be expected for the hypothetical model we described here, which involves the enzymatic activities of the two proteins tested on RNA molecules having high turnover rates. With the aim to address this Reviewer's criticism, we performed a dedicated Co-IP interaction analysis. In agreement with the hypothesis of a weak and transitory binding between the two proteins, we were unable to detect any strong interaction, while the previously characterized APE1 binding to NPM1 was perfectly reproducible. These comments have been introduced in the Revised version of the Manuscript.

Co-immunoprecipitation (CoIP) analysis on HeLa cells transfected with APE1 WT FLAG-tagged proteins with endogenous DROSHA after treatment with 1mM H₂O₂ for 15 min. Total cell extracts were immunoprecipitated with FLAG antibody and Western blot analysis was used to quantify the interaction among APE1 and DROSHA. Western blot analysis was performed on total cell extracts (left) and on immunoprecipitated material (right) with specific antibody for endogenous DROSHA and FLAG for APE1 transfected.

In addition, we additionally proved the specificity of interaction between APE and DROSHA upon H₂O₂-treatment by PLA-assay in APE1-kd cells, as also requested by Referee #2 in Q2.6. These data have been reported in Supplementary Fig. 4c,d. As expected (Supplementary Fig 4c,d), APE1 kd resulted in a significant reduction in PLA signals upon H₂O₂-treatment, paralleling the reduced interaction between APE1 and DROSHA in APE1-kd cells.

Q3.3. I'm puzzled why miR-222 was more strongly down-regulated in APE1 knockdown cells (Figure 2d), while only miR-221 was identified as a differentially expressed miRNA in cells stressed by H₂O₂ (Figure 1b). Figure 4b is even more confusing, why are mir-221 and mir-222 showing distinct behaviors? Any explanation?

A3.3. The levels of the two mature miRNAs (Fig. 2d) were measured before and after APE1-silencing. miR-222 expression resulted about 6-fold more abundant than that of miR-221 in HeLa control cells (SCR). miR-222 was significantly decreased upon APE1 silencing (siRNA). In the case of miR-221, we observed a certain trend of decrease, although not statistically significant. On the other hand, the amount of each pri-miR was increased upon APE1 silencing, as calculated from the corresponding miR to pri-miR ratios (Fig. 2e). In order to substantiate the essential role of

APE1 for the maturation process of miR-221/222, we used a recently developed APE1-ko mouse cell model (i.e. CH12F3) (Masani et al., Mol Cell. Biol., 2013). Data obtained in this case showed a direct and clear relationship between the expression of APE1 and the expression levels of mature miR-221/222 (Supplementary Fig. 2). Overall, these results indicate that, although to a different extent possibly dependent on the absolute miR expression levels, the pri-miR-221/222 processing is compromised in APE1-depleted cells.

Therefore, we modified the text accordingly (Page 10, Lines 222-229).

Regarding Figure 4b, we have now clarified this point, as follows. This experiment was performed in HeLa cells and not in APE1-kd cell clones; all RNA levels were expressed as relative to untreated controls (NT) and not as absolute values, as in Figure 2c. This point has been better clarified in the amended legend of the vertical axis of each panel. Thus, the different kinetics observed in the case of the two miRNAs, particularly once starting the release time upon H₂O₂-treatment (indicated as time 0 of release), may be ascribed to a different turnover rate of the two miRNAs. These comments have been now added in the text (Page 14, Lines 313-315).

Minor points

Q3.4. "Precursor miRNA" and "pre-miRNA" are generally used to refer to the short hairpin that is released by Drosha, not the primary transcripts. In this manuscript, the word "precursor" is used when referring to primary transcripts, which is somewhat confusing. The author should avoid this.

A3.4. We carefully corrected this improper use of the word 'precursor' all throughout the text.

Q3.5. It is unclear the figure for in vitro cleavage assay is part of Fig 3a or 3b.

A3.5. Figure 3 also describes a classical cleavage assay that is commonly used to test the enzymatic activity of APE1 on an abasic substrate. This assay is clearly described in the methodological section (paragraph entitled: AP-site Incision Assays) as well as in several published papers, which have now been indicated within the text.

Q3.6. In Figure 3b, it is unclear what OCI/AML-2 and OCI/AML-3 cell lines are and the difference between the two.

A3.6. As already explained above (see Referee #2), we have better described the difference between the AML cell clones in the Results section (Page 12, Lines 263-264) and in the Figure Legend to Fig. 3b. In particular, OCI/AML3 cells, which stably express a NPM1c+ mutant protein relocalizing to the cytoplasmic compartment, have a significant accumulation of cytoplasmic APE1 as a consequence of its interaction with the abnormal NPM1 and a consequent BER-impairment, as we previously demonstrated¹⁶. Conversely, the OCI/AML2 line represents the control cell clone expressing a wild-type NPM1 protein, which correctly accumulates within nucleoplasm and nucleoli.

Q3.7. Line 305 refers to abbreviated NSCLC. The full description of NSCLC can only be found in the Methods section.

A3.7. We have now introduced the full name in the amended text (Page 16, line 347)

Q3.8. Figures should be labeled more accurately. Figure 2C Y-axis has duplicated numbers, Figure 5C Y-axis should be "relative protein level" not "fold change", and Figure 6B Y-axis would not be "%" (I am assuming it shows the fraction of each category.), just to point out a few. In the

main text, when the authors discuss the correlation (page 14), some values are shown as “r” and others as “r²”. The statistical method used should be indicated and a consistent method should be used. Line 318, I assume that “r²=418” is incorrect.

A3.8. We apologize for the mistakes. In this amended manuscript, we have relabeled Figure 6b Y-axis as “Fraction of each PTEN score”, as suggested. We also have changed all r-square to r-value; in fact, it more accurately and originally represents the correlation between two parameters. Moreover, we also corrected the r-value to “r=0.418” in Line 360. Finally, we have indicated the Statistical Method used for this analysis (Spearman's rank correlation test).

Q3.9. Throughout the manuscript, there are sentences that I could not comprehend. Just to point out one, I could not understand what this sentence means: Page 22 line 479-482 “Whether its ability to... its importance as a druggable target for cancer therapy will be definitely demonstrated.”, for example. Also, besides the issue with the sentence structure, the last part of this sentence regarding the importance of the finding in therapeutics may be too strong, because the present study still does not directly clarify the extent to which APE1 regulates genes via regulatory non-coding RNAs, particularly in cancers.

A3.9. The manuscript has been now read and corrected by a native English speaker. In accordance, the highlighted sentence was clarified, as correctly pointed out.

Reviewers' Comments:

Reviewer #2:

Remarks to the Author:

The paper here is a revision of a manuscript that detailed a novel function for APE1, a DNA repair protein, in miRNA processing. The paper has been significantly improved, yet there remain a few additional points that require attention:

The data of Figure 3b are a little unclear. There does not appear to be any endogenous APE1 protein in the empty and N33 cell lines. Wasn't the same parental cell line being used in each case? Or was that simply variation in the effects of transient knockdown? I certainly could have missed, but I didn't find a description of how these studies were performed. Some clarification and explanation are needed, although I realize the details will not likely change the bigger picture conclusions.

I think it's important for the authors to point out that the N33 deletion mutant lacks critical localization signals and likely has impaired interactions with other proteins, both of which could participate in the cellular endpoints measured. Certainly, a reduced interaction with NPM1 is not the only defect of this mutant.

Why not include the co-IP experiments described in the rebuttal letter in the manuscript as Supplementary Data (instead of data not shown). I feel important.

While the revised paper is now written with a great level of clarity, there still exist several parts that could be improved in terms of the English grammar. Further professional editing is recommended.

Reviewer #3:

Remarks to the Author:

In the revised manuscript and response letter, the authors addressed most of the issues I raised last time. However, I still have concerns regarding the lack of biochemical evidence for the interaction between Drosha and APE1. Based on the description in the letter, "The nature of the interaction between APE1 and Drosha should be weak and transitory, as it would be expected for the hypothetical model we described here, which involves the enzymatic activities of the two proteins tested on RNA molecules having high turnover rates.", I am assuming that their concern is the transient nature of the interaction between Drosha or APE1 due to their enzymatic activities. If so, why can't the catalytically inactive mutants or the inhibitors be used?

I am still unclear about the time course results in Figure 4b. Are mir-221 and mir-222 supposed to be produced from a single polycistronic transcript? If so, can the kinetics of primary transcripts for mir-221 and mir-222 be different? I thought pri-mir-221 and pri-mir-222 should be the same molecule. Even if their kinetics can be different, why is pri-mir-221 increased at time 0 of release where supposedly cells should be able to process pri-mir-221 better than NT.

Supplementary figure S2 is convincing regarding the difference in the mature miRNA levels but lacking pri-mir-221/222 expression data. The pri-miRNA data are needed to exclude the possibility of transcriptional effects.

Point-by-point answers to Referees' requests

Reviewer #1

No additional comments were provided by this Referee.

Reviewer #2

We thank this Referee for his/her suggestions to ameliorate the quality of our Manuscript. We modified the text as suggested and as indicated below.

Major points:

Q.2.1. The data of Figure 3b are a little unclear. There does not appear to be any endogenous APE1 protein in the empty and N33 cell lines. Wasn't the same parental cell line being used in each case? Or was that simply variation in the effects of transient knockdown? I certainly could have missed, but I didn't find a description of how these studies were performed. Some clarification and explanation are needed, although I realize the details will not likely change the bigger picture conclusions. I think it's important for the authors to point out that the N33 deletion mutant lacks critical localization signals and likely has impaired interactions with other proteins, both of which could participate in the cellular endpoints measured. Certainly, a reduced interaction with NPM1 is not the only defect of this mutant.

A.2.1. For these experiments, we used the same HeLa cell clone after inducible silencing of the endogenous APE1 expression through specific shRNA (described in Vascotto et al., 2009). Further, we transiently transfected it (where is the case) with an empty-plasmid as a control or a plasmid encoding a siRNA resistant flag-tagged APE1 form, which bears two mismatches in the cDNA sequence preventing the degradation of the ectopic APE1 mRNA, while leaving the APE1 amino acid sequence unaffected. As requested, these experimental details have been better clarified in the manuscript and in the figure legend. As correctly pointed out by this Referee, the endogenous APE1 protein expressed in the empty clone is barely visible, while it is a bit more visible in APE1-transfected cells. This reduced effect of the shRNA sequences in APE1-transfected cells may be explained by the fact that the ectopic APE1-mRNA expressed sequences may act as sponges/targets for the APE1-siRNA sequences, thus reducing the effect of silencing itself.

In the case of the N Δ 33 deletion mutant, this protein form co-migrates with the endogenous one; thus, it is difficult to separate the two APE1 species in SDS-PAGE analysis. With the aim to clarify this point, we added a Western blot developed with the anti-Flag antibody in order to show the migration of the ectopic flag-tagged protein in each sample (Figure here below and reported in the new Fig. 3b).

Figure 1 Mature miR to pri-miR ratios in HeLa cell clones silenced for the endogenous APE1 expression and transiently transfected with expression plasmids for FLAG-tagged, siRNA-resistant APE1 mutants APE1^{WT}, APE1^{NΔ33}, APE1^{E96A} and APE1^{C65S}. Mature miR-221 and miR-222 levels were measured by qRT-PCR analysis, normalized to RNU44, and expressed as relative to GAPDH-normalized pri-miR-221/222. Asterisks represent a significant difference with respect to control (SCR). * P<0.05, ** P< 0.001.

Below, Western blotting analysis showing HeLa cell clones silenced for endogenous APE1 protein (*endo*) and re-expressing ectopic APE1 FLAG-tagged mutants (*ecto*). Expression of the ectopic APE1 forms was assayed by Western blotting with anti-FLAG antibody.

In any case, in accordance with this referee, we believe that overall these details do not affect at all the bigger picture conclusions we described.

Moreover, we agree with this Referee's observation that the NΔ33, lacking critical localization signals, has impaired interactions with other proteins besides a reduced interaction with NPM1, as we also previously demonstrated (Vascotto et al., 2009), both of which could participate in the cellular endpoints measured. Therefore, we added a sentence pointing out this aspect (Page 11, Lines 234-237).

Q.2.2. Why not include the co-IP experiments described in the rebuttal letter in the manuscript as Supplementary Data (instead of data not shown). I feel important.

A.2.2. As suggested, we have now included in this revised Manuscript as Supplementary Data (Supplementary Fig. 4f) the co-IP experiments described in our previous rebuttal letter.

Figure 2 Co-immunoprecipitation (CoIP) analysis on HeLa cells transfected with APE1 WT FLAG-tagged proteins with endogenous DROSHA after treatment with 1 mM H₂O₂ for 15 min. Total cell extracts were immunoprecipitated with FLAG antibody and Western blot analysis was used to quantify the interaction among APE1 and DROSHA. Western blot analysis was performed on total cell extracts (left) and on immunoprecipitated material (right) with specific antibody for endogenous DROSHA and FLAG for APE1 transfected.

Q.2.3. *While the revised paper is now written with a great level of clarity, there still exist several parts that could be improved in terms of the English grammar. Further professional editing is recommended.*

A.2.3. This manuscript has now been read and corrected for text and English grammar by a colleague from the USA (Prof. Bruce Demple, Stony Brook University, Stony Brook, NY, USA) which we acknowledge in the Acknowledgement section.

Reviewer #3

We thank this Referee's additional stimuli to increase the quality of our work. We have performed the additional experiments and analyses suggested and specific responses to the issues raised are as follows.

Major points:

Q.3.1. In the revised manuscript and response letter, the authors addressed most of the issues I raised last time. However, I still have concerns regarding the lack of biochemical evidence for the interaction between Drosha and APE1. Based on the description in the letter, "The nature of the interaction between APE1 and Drosha should be weak and transitory, as it would be expected for the hypothetical model we described here, which involves the enzymatic activities of the two proteins tested on RNA molecules having high turnover rates.", I am assuming that their concern is the transient nature of the interaction between Drosha or APE1 due to their enzymatic activities. If so, why can't the catalytically inactive mutants or the inhibitors be used?

A.3.1. We appreciate this Referee's concern. However, it is well known that the most common problem encountered in co-IP experiments is the loss of interacting partners during the washing process, which is critical for the success of the technique [Vasilescu J, Figeys D (2006) Mapping protein-protein interactions by mass spectrometry. Curr Opin Biotechnol 17: 394–399. 14. Trinkle-Mulcahy L, et al. (2008) Identifying specific protein interaction partners using quantitative mass spectrometry and bead proteomes. J Cell Biol 183: 223–239.]. Ideally, washing should be sufficiently vigorous to maximally remove nonspecifically bound proteins, while retaining co-precipitated interacting proteins on the beads. Fewer washes or use of less stringent buffers tend to identify more binding partners, but also significantly increase the number of non-specifically interacting proteins. Moreover, especially once the protein complexes involve unstable molecules, such as RNA, and particularly once the interaction imply an enzymatic activity on RNA by the interacting proteins or in the case of low-abundance binding partners, such as the case of Drosha, co-IP analysis is not the ideal technique to characterize protein-protein interactions, at least under our stringent experimental conditions. For these reasons, we chose PLA analysis to measure APE1-Drosha interaction in fixed cells, for which we provided solid demonstration of the specificity of the analysis performed, in our previous revision to our Manuscript, as acknowledged by both Referees.

As previously requested by this Referee, and already shown in our rebuttal letter to the previous revision, we performed co-IP analysis in order to give biochemical support to our PLA analysis, but we obtained negative results. We ascribed these findings to the transient nature of the interaction as well as to the possibility that RNA, mediating this interaction and due to its unstable nature, may be degraded during the washing and the preparations steps.

In the second editorial step, Referee #3 has suggested to try additional Co-IP validation of the described interaction by using the E96A APE1 catalytic mutant that, for the reasons listed above and as supported by the literature cited above, could not result in the definitive biochemical proof expected by this Referee. This difficulty could, in principle, be solved by using different chemical cross-linker reagents (such as DSP or formaldehyde). Nevertheless, the set-up of a proper biochemical protocol may eventually require a lot of time and, more importantly, may incur in the risk of artifacts. Accordingly, we consider it actually largely beyond the scope of this present manuscript, not substantially adding much to its overall scientific messages and conclusions.

Following suggestion by Referee #2, we have now included the negative results in the Supplementary material file (Supplementary Fig. 4f) and we have decided not exploring further the biochemical characterization of the APE1-Drosha interactions through other techniques than the Co-IP, leaving these data for further works.

Q.3.2. I am still unclear about the time course results in Figure 4b. Are mir-221 and mir-222 supposed to be produced from a single polycistronic transcript? If so, can the kinetics of primary

transcripts for mir-221 and mir-222 be different? I thought pri-mir-221 and pri-mir-222 should be the same molecule. Even if their kinetics can be different, why is pri-mir-221 increased at time 0 of release where supposedly cells should be able to process pri-mir-221 better than NT.

A.3.2. We thank this Referee for this keen observation that prompted us to perform additional bioinformatics analyses, which clarified this aspect and that could be of general interest for the readers.

The TaqMan® Pri-miRNA assay used in this work was designed to detect the transcript that contains the targeted stem-loop. TaqMan Pri-miRNA Assays have been designed in close proximity to each stem-loop sequence identified in the Sanger miRBase sequence repository. In each case, the assay is located within 500 nucleotides on either side of the stem-loop sequence. In some instances, when stem-loop sequences are clustered together and are transcribed on the same transcript, a TaqMan® Pri-miRNA assay targeting one of the stem-loops will also detect the other stem-loops in the cluster. Looking carefully through datasheet details, it has been reported that pri-miR-221 and pri-miR-222 have different loops, different locations and mature miRNA is also different. Therefore, all these aspects indicate that they are different pri-miRNAs detected univocally by the assay.

As correctly pointed out by this Referee, it is commonly thought that miR-221/222 belong to a polycistronic cluster giving rise to a unique pri-miRNA sequence. However, the transcriptional regulation of this cluster has not been completely elucidated, yet. In fact, data from both experimental and ENCODE analyses (Rommer et al., BMC Cancer 2013, 13:364) revealed that this cluster may be originated from different transcripts of different lengths (i.e. 5.6, 28.2 and 108.5 kb), which contain the sequences for mature miR-221/222 around 3-4 kb; no specific characterization of the promoter regions has been published yet. Our data, showing different expression levels for the two pri-miRNAs, suggested that an additional Transcription Start Site (TSS) may be present in the spacer region between the two pri-miRNA sequences.

In order to try to understand if multiple TSS were indeed present in the miR-221/222 locus, we used the atlas of promoter activities released by the FANTOM5 project (Forrest ARR, Kawaji H, Rehli M, Baillie JK, de Hoon MJL, Haberle V, Lassman T, Kulakovskiy I V, Lizio M, Itoh M, et al. 2014. A promoter-level mammalian expression atlas. *Nature* 507: 462–70.) to measure TSS and the promoter usage across a collection of over 1800 human samples. This approach allowed to define two different regions of transcription initiation, one associated with the expression of miR-221 (hg19::chrX:45605575-45606248) and a second one associated with miR-222 (hg19::chrX:45606341-45606513); see Reviewer Figure 1A, here below.

In the 1174 human samples where both miRNAs are expressed, the RLE-normalized promoter activity of miR-222 is on average 3.8-fold higher than that of miR-221 (median value = 2.8-fold) (see Reviewer Figure 1B). Moreover, there are 284 samples where only miR-222 is expressed and 74 samples where only miR-221 is expressed, and 297 samples where both samples are not expressed.

We also restricted this comparison to the HeLa cell line, as we used in this manuscript, confirming the higher promoter activity of miR-222 (average RLE-normalized expression of pri-miR222 4.4-fold higher than pri-miR221, median RLE-normalized expression of pri-miR222 2.6-fold higher); see Reviewer Figure 1C. All these data perfectly matched with our observations, which show higher expression level of pri-miR-222 than pri-miR-221 (Average expression of pri-miR-222 3.5-fold \pm 0.12 higher than pri-miR-221) in the HeLa cell line we used.

Therefore, it is possible that the difference we observed in the pri-miRNA expression levels under basal conditions (Fig. 2c) and under oxidative stress conditions (Fig. 4b) could be ascribed to the use of alternative promoters, leading to the expression of different levels of the two pri-miRNAs molecules. Further work is required to address this interesting issue that is actually far from the scope of the present manuscript.

In order to clarify this aspect that could be of general interest for the readers, we inserted a sentence of comment in the Discussion section, on page 20, line 442: “Interestingly, we noticed that the different pri-miR-221 and pri-miR-222 expression levels we measured, may be suggestive for independent expression by different promoters, as also confirmed by experimental data

obtained from the FANTOM5 project (Forrest ARR, Kawaji H, Rehli M, Baillie JK, de Hoon MJL, Haberle V, Lassman T, Kulakovskiy I V, Lizio M, Itoh M, et al. 2014. A promoter-level mammalian expression atlas. *Nature* 507: 462–70). In fact, in the 1174 human samples analysed, the RLE-normalized promoter activity of miR-222 is on average 3.8-fold higher than that of miR-221. Therefore, in addition to a commonly thought polycistronic nature, miR-221 and -222 may be independently transcribed”

Figure 3 Bioinformatics evidence of independent transcription of miR-221 and miR-222. A) Modified screenshot of the genomic region for miR-221 and miR-222 in ZENBU Genome Browser (<http://fantom.gsc.riken.jp/zenbu/gLyphs/>) showing the presence of independent different CAGE-seq peaks for the two miRNAs. B) Scatter plot of CAGE-seq peak activities for miR-221 and miR-222 in the FANTOM5 sample collection (n=1830). C) Table showing the CAGE-seq peak activities for miR-221 and miR-222 in the HeLa cells.

Concerning the kinetics analysis and the increase in miR-221 to pri-miR-221 ratio at time 0, it must be clarified that time 0 refers to the time of release upon 15 min of treatment with H₂O₂. As mentioned above, it is possible that the two miRNAs should follow different and independent kinetics of processing under basal or oxidative stress conditions. Therefore, the different kinetics observed in the case of the two miRNAs, particularly once starting the release time upon H₂O₂-treatment (indicated as time 0 of release), may be ascribed to a different turnover rate of the two miRNAs, as we already indicated in the Results section (Page 13, from line 287 on).

Q.3.3. *Supplementary figure S2 is convincing regarding the difference in the mature miRNA levels but lacking pri-mir-221/222 expression data. The pri-miRNA data are needed to exclude the possibility of transcriptional effects.*

A.3.3. As requested by this Referee, we now have performed the expression analysis of the pri-miR221/222 in the APE1 mouse knock-out cell line. Resulting data confirmed the accumulation of

both pri-miRNAs species and the concomitant reduction of the mature forms in the APE1^{-/-} mouse cell line already observed in APE1-kd human cancer cell lines (Fig. 2). These data definitively confirm that APE1-loss of expression is associated with an impairment of the maturation process of the miR-221/222, as we hypothesized in our Manuscript. These data have now been introduced in the new Supplementary Fig. 2 as ratio miR to pri-miR in accordance to the other figures of the Manuscript.

Figure 4 Mature miR-221 and miR-222 expression levels evaluated by qRT-PCR analysis of APE1-null CH12F3 cells. Total RNA was extracted from CH12F3 APE1^{Δ/+} and APE1^{Δ/Δ} cells and reverse transcribed. Histograms show the detected levels of miR-221 and miR-221

Reviewers' Comments:

Reviewer #2:

Remarks to the Author:

The authors have adequately addressed my concerns.

Reviewer #3:

Remarks to the Author:

The responses and edited manuscript satisfactorily addressed my concerns, and I support publication of the manuscript in Nature Communications.